# RETRI3D: 3D NEURAL GRAPHICS REPRESENTATION RETRIEVAL

**Yushi Guan**[1], **Daniel Kwan**[1], **Jean Sebastien Dandurand**[1], **Xi Yan**[1], **Ruofan Liang**[1],
**Yuxuan Zhang**[1], **Nilesh Jain**[2], **Nilesh Ahuja**[2], **Selvakumar Panneer**[2], **Nandita Vijaykumar**[1]

`{guanyushi, dkwan, js.dandurand, xyan, ruofan, yuxuan, nandita}@cs.toronto.edu`
`{selvakumar.panneer, nilesh.jain, nilesh.ahuja}@intel.com`

## ABSTRACT

Learnable 3D Neural Graphics Representations (3DNGRs) have emerged as promising 3D representations for reconstructing 3D scenes from 2D images. Numerous works, including Neural Radiance Fields (NeRF), 3D Gaussian Splatting (3DGS), and their variants, have significantly enhanced the quality of these representations. The ease of construction from 2D images, suitability for online viewing/sharing, and applications in game/art design downstream tasks make it a vital 3D representation, with potential creation of large numbers of such 3D models. This necessitates large data stores, local or online, to save 3D visual data in these formats. However, no existing framework enables accurate retrieval of stored 3DNGRs. In this work, we propose, Retri3D, a framework that enables accurate and efficient retrieval of 3D scenes represented as NGRs from large data stores using text queries. We introduce a novel Neural Field Artifact Analysis technique, combined with a Smart Camera Movement Module, to select clean views and navigate pre-trained 3DNGRs. These techniques enable accurate retrieval by selecting the best viewing directions in the 3D scene for high-quality visual feature embeddings. We demonstrate that Retri3D is compatible with any NGR representation. On the LERF and ScanNet++ datasets, we show significant improvement in retrieval accuracy compared to existing techniques, while being orders of magnitude faster and storage efficient.

## 1 INTRODUCTION

A radiance field maps 3D spatial locations and 2D viewing directions to a radiance value, typically encoding the color and intensity (Pharr & Humphreys, 2016). It has recently been combined with learnable representations, such as neural networks and trainable 3D Gaussians, revolutionizing novel view synthesis tasks (Mildenhall et al., 2020; Kerbl et al., 2023). We refer to these solutions, which learn radiance fields for novel view synthesis, as 3D Neural Graphics Representations (3DNGRs). Traditional 3D formats, such as meshes, voxels, and point clouds, explicitly define the geometry of the 3D data. In contrast, NGRs use differentiable rendering to learn the 3D scenes from sparse sets of 2D images. This approach has many advantages including compact storage (Mildenhall et al., 2020), continuous and unbounded representation (Barron et al., 2021; Kerbl et al., 2023), and high-quality novel view synthesis (Liang et al., 2023). Additionally, NGRs can be integrated with AI content generation pipelines (Poole et al., 2022) with many applications in 3D art and game design (Saito et al., 2023; Gu et al., 2023; Unity Technologies, 2024; NVIDIA Corporation, 2024). The ease of constructing NGRs from casually captured 2D images has resulted in an abundance of diverse 3D scenes available for viewing, sharing, and downloading online (Polycam, 2024; Luma AI, 2024).

Despite the abundance of NGR assets, a framework for storing and retrieving 3D scenes represented as NGRs from a data store has not yet been developed. Many existing works tackle the storage and retrieval of images and traditional 3D formats (meshes, point clouds, voxel grids) (Tan et al., 2021; Lahav & Tal, 2020; Sanghi, 2020; Wang et al., 2018). Existing NeRF semantic understanding solutions are performed on a scene-by-scene basis and cannot be easily scaled to hundreds of scenes as a general retrieval solution (Kerr et al., 2023; Qin et al., 2023).

In this work, we identify three main challenges in enabling efficient retrieval of NGR assets in a data store: **First**, the representation format is highly heterogeneous. A wide range of NGR formats exist, for example, NeRF, 3DGS, and their variants. In addition, new formats are continuously being developed. It is critical for a retrieval system to be compatible with any pre-trained NGR, ensuring that existing 3D assets do not become obsolete and that new NGR formats can be seamlessly added to the

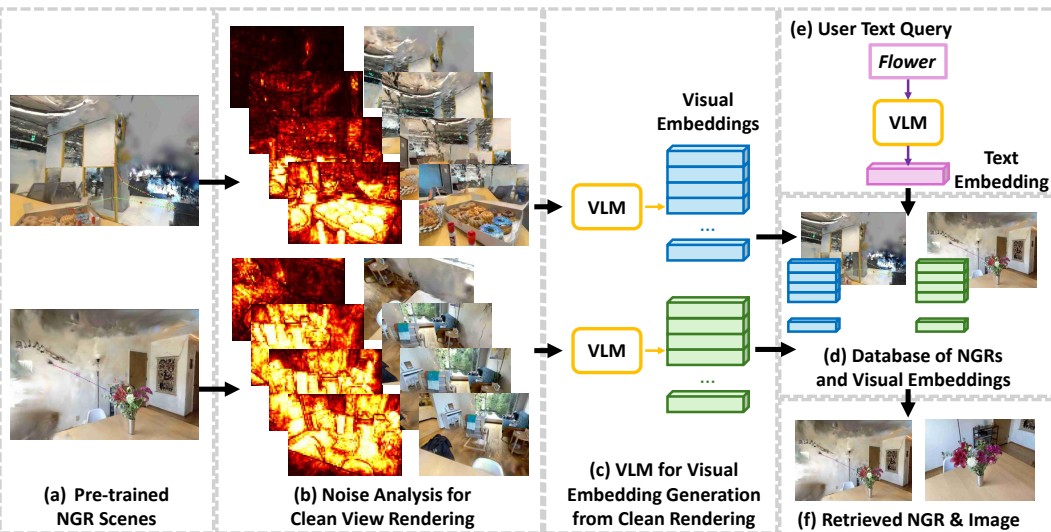

Figure 1: Overview of Retri3D: (a) Retri3D enables the storage and retrieval of any neural graphics representation (NGR). (b) We perform noise analysis and viewpoint selection to select high-quality artifact-free views (See Section 3.3 and 3.4). Videos of rendered trajectories are provided in the supplementary material. (c) Visual embeddings for retrieval are generated using a pre-trained Vision-Language Model (VLM). (d) Both NGRs and visual embeddings are stored in the database. (e) Given a user query, we use the same VLM to generate a corresponding text embedding. (f) Retrieval of the relevant scene is based on the highest cosine similarity between the text and visual embeddings, returning the NGR and optionally, a rendered image, to the user.

data store. **Second**, 3D scenes exhibit a high degree of diversity and complexity. Since NGRs allow 3D scenes to be easily constructed from casually captured 2D images, a retrieval solution must be compatible with any semantic concepts or sentence-like descriptions. Unlike traditional 3D retrieval solutions that focus on isolated 3D objects with a single concept, 3D scenes contain numerous potential views, each with multiple concepts. This necessitates a solution capable of extracting different visual features from different parts of the scene. Moreover, the solution should efficiently extract relatively compact embeddings to minimize storage and computational overhead. **Third**, the presence of noise and floater artifacts in NGRs makes visual feature extraction from the scenes much more challenging. These artifacts are primarily due to incomplete scene coverage during capture but are also affected by factors such as image blurs, depth, and lighting ambiguities. For a 3D scene, artifacts can significantly affect rendered 2D images from randomly selected viewpoints, even with methods specifically designed to reduce floaters and artifacts in NGRs (Philip & Deschaintre, 2023). The pre-trained NGRs are typically stored by itself without the images used for training (Polycam, 2024), making the selection of good viewpoints for understanding the scene crucial. We demonstrate that these floaters severely impact visual feature extraction of the scenes, highlighting the need for a solution that can efficiently detect and identify floater-free viewpoints in any pre-trained NGR.

In this work, we propose Retri3D, the first framework that addresses these challenges to enable accurate and efficient retrieval of NGRs of any format using text queries. To enable generality across various NGR formats, we propose to first render images from the 3DNGRs and then generate visual semantic embeddings from the rendered images. We demonstrate that selecting random viewpoints for rendering images from the 3D representations leads to artifacts, severely impacting the quality of visual embeddings and, thus, the accuracy of the retrieval. To address this, we introduce a novel *Neural Graphics Noise Analysis* technique, which can identify artifacts in an RGB image. We compare with state-of-the-art NeRF uncertainty measurement work, and demonstrate that our solution identifies the RGB pixel error in the rendered images with higher accuracy (Goli et al., 2023). Coupled with a *Smart Camera Movement Module*, our system navigates the camera within the 3D scene to viewpoints that ensure high-quality rendering by avoiding floater artifacts. We show that our solution significantly improves the efficiency and accuracy of retrieval. Finally, we leverage recent advances in pre-trained Vision Language Models (VLMs) to convert high-quality rendered images into compact visual embeddings in an efficient manner. Our system utilizes the zero-shot capabilities of pre-trained VLMs, allowing it to easily benefit from potential future improvements in VLMs.

We demonstrate the effectiveness of Retri3D using the LERF and ScanNet++ datasets (Kerr et al., 2023; Yeshwanth et al., 2023). For the LERF dataset, we show that our system can extract visual features from highly varied 3D scenes with complex details. Compared with adapting existing NGR semantic analysis solutions for retrieval, we demonstrate our system has higher accuracy while being more general and orders of magnitude faster and storage-efficient. On the ScanNet++ dataset, we show that our solution scales to hundreds of scenes, performing a scene lookup in 1 millisecond while only requiring under 20 seconds for visual feature extraction of the scene during insertion.

This work makes the following contributions: (i) We present the first end-to-end framework for accurate and efficient retrieval of 3D scenes represented as NGRs using text queries. (ii) We propose a novel technique to detect NGR noise through rendered RGB images and pre-trained VLMs, achieving superior RGB error measurement accuracy over existing methods. (iii) We develop a solution for rapid camera movement to artifact-free viewpoints for high-quality embedding extraction. (iv) Through LERF and ScanNet++ datasets, we show that Retri3D achieves efficient, scalable, and high-accuracy 3D scene retrieval.

## 2 RELATED WORK

To our knowledge, this is the first work that tackles the accurate and efficient retrieval of 3D scenes represented as NGRs. Our proposed work leverages advances in pre-trained Vision-Language Models (VLMs). It is inspired by research in related fields such as traditional 3D object retrieval and 3D semantic understanding. We now describe the most relevant works in this section.

### 2.1 LEARNABLE 3D NEURAL GRAPHICS REPRESENTATIONS

In recent years, neural graphics methods have demonstrated great advances in 3D reconstruction and novel view synthesis, with innovative methods, such as neural radiance fields (NeRFs) (Mildenhall et al., 2020) and 3D Gaussian splatting (3DGS) (Kerbl et al., 2023), being proposed.

NeRF-based methods (Barron et al., 2021; Müller et al., 2022; Chen et al., 2022; Yu et al., 2021) represent the 3D scene as spatial volumes with neural parameters (e.g., MLPs, feature grids, and hashtables, etc.), optimized through volumetric rendering. 3DGS (Kerbl et al., 2023) uses trainable 3D Gaussians with alpha-blending rasterization to replace NeRF's volumetric rendering, achieving faster training and inference speed.

There are other types of 3D neural representations, such as neural signed distance fields (Wang et al., 2021; Yariv et al., 2021) and differentiable meshes (Shen et al., 2021b; Munkberg et al., 2021), also demonstrating great performance in 3D reconstruction tasks. Given different NGR formats, formulating a completely general solution for analyzing the learned parameters is extremely challenging. With Retri3D, we propose to use RGB image renderings for embedding generation to enable a fully general solution.

### 2.2 PRE-TRAINED VISION LANGUAGE MODELS

The introduction of large-scale text-image contrastive learning by CLIP has significantly improved the performance of pre-trained Vision Language Models (VLMs) (Radford et al., 2021). These large models are capable of understanding and processing a wide range of natural language inputs and images, making them highly versatile. Recent works apply similar large-scale pre-training approaches to tasks such as object detection and (semantic) segmentation with open-vocabulary (Kirillov et al., 2023; Liu et al., 2023b; Zhang et al., 2023a; Zou et al., 2022; 2023a). In this work, we leverage different pre-trained VLMs (XDecoder and OpenCLIP) to demonstrate our solution's applicability when leveraging different VLMs (Zou et al., 2022; Cherti et al., 2022). We use the VLMs in two ways: (i) we use their visual and textual embeddings for text-scene retrieval; (ii) we use their activations to determine if a region contains NGR noise artifacts or not. To the best of our knowledge, we are the first to demonstrate noise analysis for NGR can be performed with pre-trained VLM activation features. Note that for all VLMs, We do not perform fine-tuning to demonstrate the noise analysis and retrieval performance achievable through zero-shot learning, thereby allowing us to generalize the performance to unseen scenes and leverage future advancements in VLMs.

### 2.3 TRADITIONAL 3D DATA RETRIEVAL

There is extensive work on the retrieval of 3D models represented in traditional explicit formats such as point clouds (Sanghi, 2020; Qi et al., 2016; 2017; Qian et al., 2022; Zhao et al., 2022), voxel grids (Wang et al., 2017; 2018), and meshes (Hanocka et al., 2019; Mitchel et al., 2021; Lahav & Tal, 2020). Another popular method involves converting 3D models into multi-view images for

retrieval (Jiang et al., 2019; Hamdi et al., 2020; Wei et al., 2020; He et al., 2018; Yavartanoo et al., 2018). One common property of these methods is their focus on the retrieval of isolated 3D objects, e.g., ModelNet (Wu et al., 2014) and ShapeNet (Chang et al., 2015). Prior works leverage deep learning methods to convert the 3D model into a single embedding, with retrieval performed based on embedding similarity. In contrast, 3D scenes encompass multiple semantic concepts/objects and cannot be easily represented as a single embedding (Peng et al., 2022; Zhang et al., 2023b; Kerr et al., 2023; Qin et al., 2023; Guan et al., 2025).

## 2.4 3D NEURAL GRAPHICS REPRESENTATIONS SEMANTIC UNDERSTANDING

Prior works integrate semantic analysis on the 2D images into NGR training. These methods often modify the NGR architecture to jointly train semantics with RGB components (Vora et al., 2021; Zhi et al., 2021; Xu et al., 2023; Liu et al., 2023a; Ye et al., 2022; Cheng et al., 2023; Fu et al., 2023; Siddiqui et al., 2022; Bhalgat et al., 2023; Hu et al., 2023; Mazur et al., 2022; Shafiullah et al., 2022; Li et al., 2022; Liu et al., 2023c; Kerr et al., 2023; Qin et al., 2023), which results in non-general, architecture-specific solutions and increased computational and storage overhead (Liu et al., 2023c; Kerr et al., 2023; Qin et al., 2023). Semantic understanding in NGRs is crucial for our work, as it is necessary for scene retrieval. Existing works can be categorized into two groups: (i) NGR object annotation and detection, and (ii) visual feature embedding. The annotation and detection methods usually apply to a set of predefined object categories (Vora et al., 2021; Zhi et al., 2021; Xu et al., 2023; Liu et al., 2023a; Ye et al., 2022; Cheng et al., 2023), which lacks rich semantic information and limits the retrieval to specific object sets.

Most relevant to our work are methods that embed visual feature embeddings into NGRs, allowing visual embeddings to be rendered and compared with any text embedding post-training. LERF (Kerr et al., 2023) and LangSplat (Qin et al., 2023) generate visual embeddings during NGR training, injecting them into NeRF and 3DGS, respectively. LERF uses multi-scale CLIP embeddings, while LangSplat segments the scene into different objects and generates embeddings for each. At test time, these visual embeddings can be rendered through volumetric rendering (LERF) or rasterization (LangSplat), similar to RGB rendering. Liu et al. (Liu et al., 2023c) proposed a similar approach that is limited to forward-facing scenes. Rashid et al. (Rashid et al.) applied LERF to a single changing scene. N2F2 (Bhalgat et al., 2024) follows LERF's methodology but explicitly encodes multi-scale visual embeddings. LEGaussians adopts the same visual embedding method as LERF but records embeddings using Gaussians instead of a NeRF field. TIGER (Xu et al., 2024) employs a visual embedding similar to LangSplat and further introduces scene editability. ConDense (Zhang et al., 2024) proposes training a 3D transformer to generate the visual embedding field, but it requires thousands of GPU hours to train a 3D transformer that is limited to a specific NGR type. ConDense also uses SuperPoint (DeTone et al., 2017) to sparsify the dense visual feature rendered from the visual embedding field, but only applicable to the visual features from training views. Other works embed visual embeddings into point clouds (Peng et al., 2022; Zhang et al., 2023b).

We adapt LERF and LangSplat methods for scene retrieval because they are open-sourced, have settings closest to our work, and differ in both visual embedding generation and encoding methods. Compared to our approach, LERF and LangSplat incur significant training and rendering overhead and generate orders of magnitude more embeddings, which negatively impact storage and retrieval speed. These limitations similarly apply to other works using similar methods. We discuss the implications of these characteristics for retrieval solutions in more detail in Appendix B.9.

## 3 METHOD

### 3.1 OVERVIEW

Retri3D retrieves a learned 3D radiance field given user text queries, offering compatibility with any pre-trained NGR format. The text query can be simple object labels or sentence descriptions. We visualize our solution in Fig 1. We assume the pre-trained scenes are shared as is, containing no information about the original training images or poses, which is typical for online sharing platforms (Polycam, 2024). In Section 3.2, we discuss how to generate diverse but compact embeddings from images. In Section 3.3, we propose a new method for detecting NGR noise and artifacts based on images only, when combined with the Smart Camera Movement Module (Section 3.4), Retri3D selects high-quality renderings to improve retrieval accuracy.

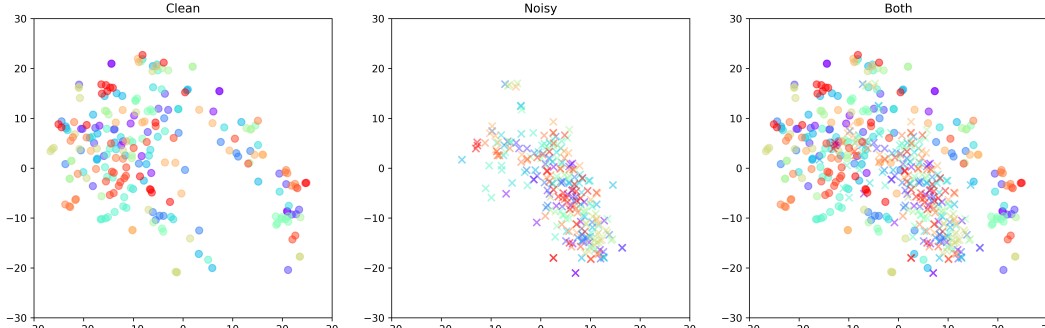

Figure 2: 2D t-SNE visualization of clean (O) and noisy (X) pixel from XDecoder. Different colors indicate different LERF scenes. The t-SNE was computed jointly using clean and noise features. They are displayed separately for clarity. We make the following observations: (1) The noise features are much more tightly clustered compared to the clean features; (2) A small amount of noise features intrude into the clean features, but they constitute a small portion and have minimal impact on the learned Gaussian; (3) The noise features from one scene are closer to the noise features from another scene, compared to clean features from their own scene.

## 3.2 VISUAL EMBEDDING GENERATION

Existing NGR semantic analysis methods require extensive training to integrate an NGR with a visual semantic field, typically represented as a NeRF or 3D Gaussian blobs. This approach necessitates manual engineering of the visual semantic field for each NGR format. Additionally, rendering the visual semantic field introduces significant computational and storage overhead, making these methods unsuitable for general-purpose NGR databases.

To address this, we propose a general solution compatible with any pre-trained NGRs capable of rendering RGB images. Our method leverages a VLM to generate visual embeddings, summarizing the image or segments of the image from NGR renderings. These embeddings are stored in the database upon acquiring the NGRs as is. During retrieval, using the same VLM, we create a text embedding from user queries and retrieve the visual embedding (along with the associated scene and image) that has the highest cosine similarity. We evaluate our approach with both an older VLM (CLIP) and a more recent VLM (XDecoder) to demonstrate its flexibility. For a detailed description of the VLMs and database indexing, please refer to Appendix A.1.

## 3.3 NEURAL GRAPHICS NOISE ANALYSIS

The existence of noise and floater artifacts in NGRs has been a longstanding problem observed by many existing works in NGR (Mildenhall et al., 2020; Warburg et al., 2023; Goli et al., 2023). In this section, we introduce Neural Graphics Noise Analysis, a method for distinguishing noise (and floater artifact) regions of NGR renderings from clean regions. The goal is to use the detected noise to redirect cameras to cleaner viewpoints to generate high-quality visual embeddings for improved retrieval accuracy. We refer to noise and floater artifacts collectively as noise below.

We observe that the activation features of pre-trained VLM exhibit the ability to separate noise regions from clean regions. In Figure 2, we show t-SNE plots of clean region features vs. noise region features generated from a VLM. Figure 2 demonstrates that the pixel-wise[1] noise features from different NGR scenes are tightly clustered, and well-separated from the clean features. To obtain these features, we pass a rendered image $I \in \mathbb{R}^{3 \times H \times W}$ through the visual encoder of a pre-trained VLM. We obtain an intermediate activation map $A = \Phi'(I; \phi')$, where $\Phi'$ is a section of the visual encoder and $A \in \mathbb{R}^{c \times h \times w}$ is the activation with features of length $c$. Our solution is compatible with different VLMs and VLM layers, see Appendix A.2.

To train a multivariate Gaussian distribution to represent the noise, we collect $K$ training (tr) activation features $\{a_{\text{tr},i}\}_{i=1}^{K}$ from random viewpoint renderings' activations to represent noise features. The renderings can be obtained from NGRs to be stored in the database, or other separate NGR scenes. We compute the mean $\mu$ and covariance $\Sigma$ of these features to fit a multivariate Gaussian distribution:

---

[1] By pixel-wise, we refer to the pixels of the activation map. This activation map can be downsampled from the original image in spatial resolution.

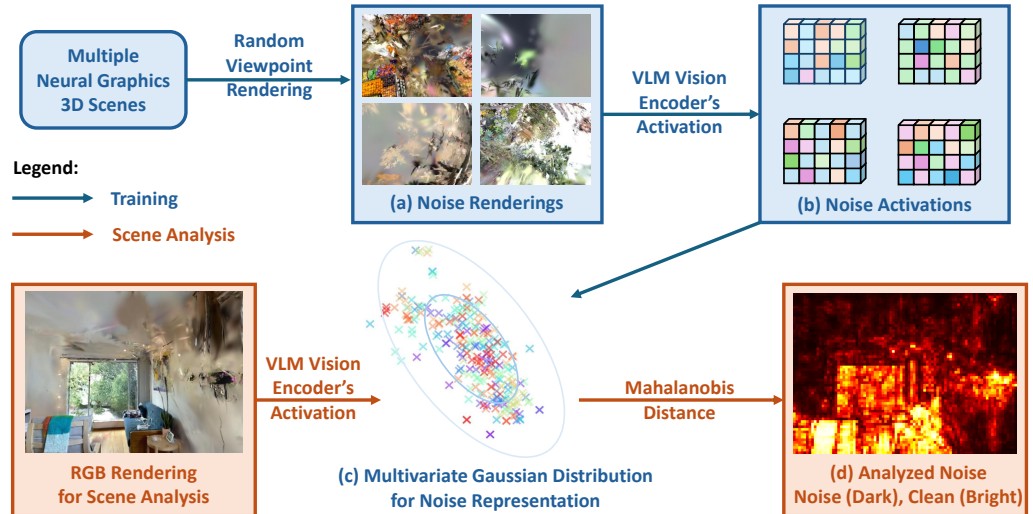

Figure 3: Noise analysis training and inference process. (a) We generate noisy images, which can be done simply through rendering random viewpoints in pre-trained NGR scenes. Some content can be noise-free, but they constitute only small portions of the images. (b) Using a pre-trained VLM's vision encoder, we generate the pixel-wise activation features. (c) We train a Multivariate Gaussian distribution to represent the noise features. (d) During inference, we calculate Mahalanobis distance of a RGB rendering's activations to the trained noise Gaussian to produce a noise map.

$$\boldsymbol{\mu} = \frac{1}{K}\sum_{i=1}^{K} \boldsymbol{a}_{\text{tr},i}, \quad \boldsymbol{\Sigma} = \frac{1}{K-1}\sum_{i=1}^{K}(\boldsymbol{a}_{\text{tr},i}-\boldsymbol{\mu})(\boldsymbol{a}_{\text{tr},i}-\boldsymbol{\mu})^{\top} + \epsilon\boldsymbol{I} \tag{1}$$

where $\epsilon$ is a small regularization constant to ensure $\boldsymbol{\Sigma}$ is invertible. For an activation feature $\boldsymbol{a}_{\text{an}}$ during the analysis of an NGR scene, we calculate the Mahalanobis distance $d_m$ to the Gaussian:

$$d_m = \sqrt{(\boldsymbol{a}_{\text{an}}-\boldsymbol{\mu})^{\top}\boldsymbol{\Sigma}^{-1}(\boldsymbol{a}_{\text{an}}-\boldsymbol{\mu})} \tag{2}$$

We then compute a normalized clean score $s$ using a sigmoid function $\sigma$:

$$s = \sigma\left(\alpha\left(d_m - c\right)\right), \quad \sigma(x) = \frac{1}{1+e^{-x}} \tag{3}$$

where $\alpha$ is a scaling factor and $c$ is a cutoff value. Pixels with larger Mahalanobis distances are considered cleaner (higher $s$), while those with smaller distances are considered noisier (lower $s$).

Note that Existing NGR works' visualizations typically use viewpoints close to the training points, resulting in images that appear clean. We explain this further in Appendix C.5. Since noisy regions dominate random viewpoints, we treat all regions from these renderings as "noise".

Our method offers several key advantages. First, it demonstrates strong generalizability: noise features remain consistent across different scenes due to shared rendering artifacts, allowing our Gaussian model, trained on random renderings, to generalize effectively to unseen scenes. We also observe similarities in noise features across different models. We further discuss the similarity of noise across models, and its impact on noise analysis in C.6. Second, it operates without requiring training image viewpoints or clean pixel features, making the approach applicable even when such data is unavailable. Third, it has low computational overhead, adding minimal processing time to the activation feature generation during VLM inference. Lastly, it is highly versatile and compatible with various pre-trained Visual Language Models (VLMs); we showcase its performance using XDecoder and CLIP.

Depending on the scenario, our method can be applied in two main approaches. The first is self-supervised noise analysis, where the Gaussian distribution is trained on random viewpoint renderings from a set of scenes; and subsequently used to analyze new renderings from the same set of scenes. We demonstrate this approach in our LERF dataset retrieval results. The second approach is cross-scene transferred noise analysis, where the Gaussian model is trained on pre-existing scenes and then applied to new, unseen scenes, leveraging the consistent nature of noise features across different environments. This method is demonstrated in our ScanNet++ retrieval results.

| **RGB** | **Clean Score** | **Next Viewpoint Selection** |
|---|---|---|

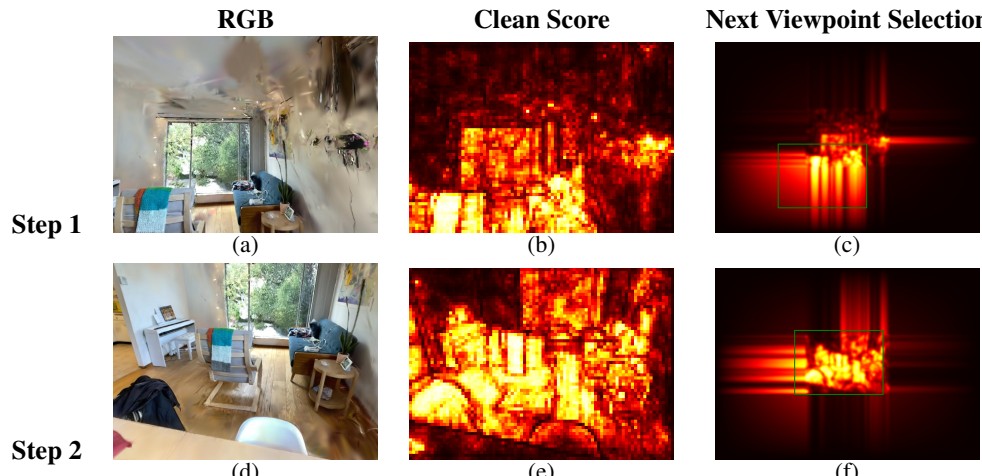

Figure 4: Application of our noise analysis method. (a) Starting from a random viewpoint rendering; (b) Noise analysis creates a clean score map highlighting clean (bright) and noisy (dark) regions; (c) Based on the noise analysis, the best next viewpoint is selected (visualized as a green box); (d) The rendering from the new viewpoint shows reduced noise. This process can be iterated multiple times.

We provide additional noise analysis experiments with different VLM layers in Appendix A.2. We demonstrate the superiority of our Noise Analysis method in identifying RGB error in rendered images compared to state-of-the-art NeRF uncertainty estimation methods in Appendix A.5. Another potential way to distinguish noise from clean pixels is to fine-tune a VLM to classify noise from non-noise. We found such a method often treats the image content with high-frequency details as noise. We provide further analysis in Appendix A.6.

### 3.4 SMART CAMERA MOVEMENT MODULE

We propose the Smart Camera Movement Module (SCMM), which discovers clean viewpoints for rendering by iteratively redirecting the camera to less noisy views. The clean views enable higher-quality visual embeddings from VLMs to improve the retrieval performance. SCMM is achieved by moving the camera towards the less noisy pixels as predicted by our noise analysis algorithm. We provide an example of such a camera movement in Figure 4.

The SCMM moves the camera to a cleaner view by estimating the noise level beyond the current field of view. We apply an iterative "render, noise analysis, camera movement" process to refine the camera viewpoint. Given a rendering, the clean score for the entire rendering is represented as $S$, where the clean score per pixel is $s_{j,i}$:

$$S = \{s_{j,i} \mid 1 \leq j \leq h, 1 \leq i \leq w\} \tag{4}$$

We estimate the clean score around the current field of view by extending it beyond the current image border. We apply a linear-decay edge-padding to the current clean score:

$$S' = \{s'_{j',i'} \mid -h \leq j' \leq 2h, -w \leq i' \leq 2w\} \tag{5}$$

$$s'_{j',i'} = \begin{cases} \left(\frac{|i'|}{w}\right) s^*_{j',1} & \text{if } 1 \leq j' \leq h, -w \leq i' \leq 0 \quad \text{(left pad)} \\ \left(\frac{|j'|}{h}\right) s^*_{1,i'} & \text{if } -h \leq j' \leq 0, 1 \leq i' \leq w \quad \text{(top pad)} \\ \left(\frac{|i'|}{w}\right)\left(\frac{|j'|}{h}\right) s^*_{1,1} & \text{if } -h \leq j' \leq 0, -w \leq i' \leq 0 \quad \text{(top left pad)} \\ \vdots & \text{(others)} \end{cases} \tag{6}$$

where $s^*$ is an pixel of the Gaussian blurred version $S^*$ of the original clean score map $S$. Given a blue kernel $G$ with std $\sigma$ and radius $k$, the Gaussian blurred clean score map $S^*$ is given by:

$$S^*_{j,i} = \sum_{m=-k}^{k} \sum_{n=-k}^{k} G(m,n) s_{j-m,i-n} = (S * G)_{j,i}, \quad G(x,y) = \frac{1}{2\pi\sigma^2} \exp\left(-\frac{x^2 + y^2}{2\sigma^2}\right) \tag{7}$$

The Gaussian blur is applied to reduce the effect of individual edge and corner pixels' clean score on the estimated clean score beyond the current view. Lastly, we efficiently find the maximum sum submatrix (Figure 4(c)(f), green box), representing the highest expected clean score. Using cumulative sum arrays, finding the submatrix only has a time complexity of $O(hw)$, linear w.r.t. the

| # | Object Label | | | LLaVA Caption | | |
|---|---|---|---|---|---|---|
| | Viewpoint | | | Viewpoint | | |
| Img. | Training | SCMM | Random | Training | SCMM | Random |
| 1 | **57.89** | 30.08 (-27.81) | 17.29 (-40.60) | **45.12** | 32.97 (-12.15) | 15.28 (-29.84) |
| 5 | **68.42** | 64.66 ( -3.76) | 18.80 (-49.62) | 55.07 | **57.17** (+2.10) | 23.58 (-31.49) |
| 10 | **78.95** | 67.67 (-11.28) | 24.81 (-54.14) | **67.07** | 61.23 (-5.84) | 28.92 (-38.15) |
| 20 | **83.46** | 73.68 ( -9.78) | 34.59 (-48.87) | **72.41** | 63.25 (-9.16) | 33.01 (-39.40) |
| 50 | **84.21** | 77.69 ( -6.52) | 49.62 (-34.59) | **71.80** | 65.82 (-5.98) | 35.58 (-36.22) |
| 100 | **84.95** | 80.02 ( -4.93) | 57.89 (-27.06) | **73.67** | 70.48 (-3.19) | 43.04 (-30.63) |

Table 1: Retrieval Accuracy (P@1) for Splatfacto Model on LERF Dataset

spatial resolution. The camera center is then rotated to point at the center of this new maximum sum submatrix.

We observe that renderings at a lower resolution can still be robust for noise analysis and the SCMM process. Therefore, to generate high-quality visual embeddings for retrieval, we propose to perform SCMM with low-resolution renderings to discover the clean viewpoint first. Then, we perform high-resolution rendering and visual embedding generation at a discovered clean viewpoint. See Appendix A.4 for details.

We discussed the application of SCMM for camera viewpoint rotation. However, by rendering sets of images that vary in depth (in the current viewpoint's coordinate), we can find the next view with the highest expected clean score in 3D. This allows our camera to both translate and rotate in 3D and move in a trajectory that iteratively points to cleaner fields. We provide further discussion and visualization in Appendix A.3.

## 4    EXPERIMENTS AND RESULTS

We perform experiments with LERF (13 scenes) and ScanNet++ (280 scenes) datasets (Kerr et al., 2023; Yeshwanth et al., 2023). We use either the object labels that came with the dataset or queries generated by the LLaVA model (using the prompt: *Describe the image in detail in one sentence.*) (Liu et al., 2023b). A retrieval is successful if the retrieved scene contains the object, or is used to generate the caption. We report retrieval accuracy P@k=1,5,10, corresponding to whether the top k retrievals contain a correct scene. Due to space limitations, we focus our results on *XDecoder + LERF dataset + Splatfacto model* in the main paper. We provide more experiments details and results including the testing platform, different/mixed NGR, and different VLMs in Appendix B.

### 4.1    COMPARISON BETWEEN DIFFERENT RENDERINGS

In this section, we compare retrieval accuracy between different rendering viewpoints. As the retrieval accuracy significantly increases as more images are used, we report accuracy for different numbers of renderings. Training Viewpoint Render is the best scenario where there is knowledge of the training camera poses, creating the highest quality renderings. However, we do not assume the training poses are available and use it as a best-case comparison. SCMM-chosen camera viewpoints utilizes our novel solution SCMM detailed in Section 3.4 to choose high-quality viewpoints for clean renderings. Random Viewpoint renders from a randomly selected viewpoint.

As shown in Table 1, the retrieval accuracy for SCMM is only a few percent lower than the retrieval accuracy with Training Viewpoint. Compared to Random Viewpoint, our solution has more than 20% higher accuracy in most cases. The accuracy differences hold for both object label text query and LLaVA caption sentence query.

### 4.2    COMPARISON WITH BASELINES

In Table 2, we compare the retrieval accuracy of our method with the LangSplat and LERF methods. As LangSplat and LERF cannot select viewpoints smartly, we render LERF and LangSplat at ***Training Camera Viewpoints***. At 100 rendered images, comparing our solution with LangSplat, we achieve 5.12% higher object label retrieval accuracy if training pose is also available to us, or 0.19% if not. LangSplat encodes visual embeddings by first segmenting objects in the scene and then using the CLIP visual encoder to encode objects individually. As a result, the embeddings are poor matches for image-level sentence descriptions. Our solution has about 20% higher accuracy than LangSplat.

The LERF model's embedding is particularly ineffective for cross-scene retrieval (explained further in Appendix B.10). LERF, trained from multi-scale CLIP image embedding (without segmentation), is very uniform and can not distinguish different objects effectively.

| | Object Label | | | | LLaVA Caption | | | |
|---|---|---|---|---|---|---|---|---|
| | Ours | | Baselines | | Ours | | Baselines | |
| # Img. | Training | SCMM | LangSplat | LERF | Training | SCMM | LangSplat | LERF |
| 1 | **57.89** | 30.08 | 44.73 | 16.32 | **45.12** | 32.97 | 18.25 | 17.63 |
| 5 | **68.42** | 64.66 | 61.16 | 16.53 | 55.07 | **57.17** | 36.62 | 19.23 |
| 10 | **78.95** | 67.67 | 62.57 | 16.67 | **67.07** | 61.23 | 42.45 | 20.58 |
| 20 | **83.46** | 73.68 | 74.21 | 18.87 | **72.41** | 63.25 | 46.84 | 26.96 |
| 50 | **84.21** | 77.69 | 76.36 | 23.45 | **71.80** | 65.82 | 49.26 | 31.48 |
| 100 | **84.95** | 80.02 | 79.83 | 26.24 | **73.67** | 70.48 | 51.92 | 36.89 |

Table 2: Retrieval Accuracy (P@1) for Different Models on LERF Dataset

| | | Ours | | Baselines | |
|---|---|---|---|---|---|
| Stage | Action or Storage | SplatFacto | Nerfacto | LangSplat | LERF |
| NGR Training | Train Time (min) | **6.82** | 8.24 | 90.5 | 40.1 |
| | Model Size (MB) | 478.49 | **176.02** | 958.67 | 1282.4 |
| NGR Analysis | Generate Visual Emb. (s) | **17.25** | 19.23 | 152.5 | 53.6 |
| Database & Retrieval | Embedding Size | **20.58MB** | **20.58MB** | 18.78GB | 225.4GB |
| | Retrieval Time (s) | $5e^{-5}$ | $5e^{-5}$ | $1e^{-3}$ | 17 |

Table 3: Speed and storage comparison with 50 rendered images for a LERF scene

## 4.3 COMPUTATION AND STORAGE EFFICIENCY

In this section, we compare the speed and storage for different stages of the retrieval process with LERF and LangSplat. As we use pre-trained NGRs as is, there is little overhead for training or storing the NGRs. Our retrieval speed is achieved through using FAISS's implementation of inverted file index *IVF1024* for efficient highest cosine similarity match (Douze et al., 2024). *IVF1024* was also used for LangSplat and LERF. As shown in Table 3, LERF and LangSplat have orders of magnitude overhead in terms of speed and storage at multiple stages. This largely results from needing to train a NGR to encode the visual embeddings, as well as the costly dense visual embedding rendering from the trained NGR. We explain this further in Appendix B.9.

## 4.4 RETRIEVAL FROM SCANNET++

Our solution easily scales to hundreds of pre-trained NGR scenes, we demonstrate through testing on the ScanNet++ dataset. We use the 280 ScanNet++ scenes with semantic labels. As the many different ScanNet++ scenes contain similar objects (such as Sofa from different living rooms). We measure the retrieval accuracy based on whether the retrieved scene contains the object label. As demonstrated in the table, ours still has a high degree of accuracy for retrieval even with hundreds of scenes. Using the *IVF1024* index, the measured retrieval speed per query is $6e^{-4}$ second. We show more results with LLaVA caption and comparison with LangSplat in Appendix B.7.

## 4.5 SCENE COVERAGE METRIC AND RESULT

To evaluate 3D scene coverage using rendered images, we collect average coverage statistics for Nerfacto across 13 LERF scenes. Utilizing Nerfacto's spatially contracted space, we partition it into a $64 \times 64 \times 64$ voxel grid and measure orientation based on the angle between camera rays and voxel faces. A voxel is considered covered if any rendering ray passes through it before reaching the estimated depth. Details are provided in Appendix A.7.

The %Grid in Table 5 shows the scene coverage (a voxel is also considered covered if the density is greater than 0.5). The %Train shows that, for voxels covered by Training, the percentage of those observed by SCMM or Random. We make two important observations. First, Training has low coverage of the scene, which explains the existence of significant noise in the NGRs. Such a low coverage is often due to the continuous video trajectory, which is how real-world NeRF datasets are usually captured. Second, SCMM provides both high coverage of the scene (%Grid), and a high coverage of the scene section that was observed by the training images (%Train). This indicates that SCMM rendering tends to favor viewpoints similar to those of the training images, enhancing the capture of semantic content present in the original video and improving the rendering quality for better retrieval.

| # Img. | Training | SCMM | Random |
|---|---|---|---|
| 10 | **41.62** | 39.63 (-1.99) | 18.39 (-23.23) |
| 20 | **50.06** | 47.34 (-2.72) | 23.58 (-26.48) |
| 50 | **58.23** | 54.39 (-3.84) | 27.93 (-30.30) |
| 100 | **65.03** | 64.18 (-0.85) | 29.04 (-35.99) |

Table 4: Retrieval Accuracy (P@1) for Splatfacto Model with Object Labels for ScanNet++

| | Training | | SCMM | | | | Random | | | |
|---|---|---|---|---|---|---|---|---|---|---|
| # Img. | %Grid | | %Grid | | %Train | | %Grid | | %Train | |
| | **xyz** | **w/$\theta$** | **xyz** | **w/$\theta$** | **xyz** | **w/$\theta$** | **xyz** | **w/$\theta$** | **xyz** | **w/$\theta$** |
| 0 | 9.9 | 9.9 | 9.9 | 9.9 | 0.0 | 0.0 | 9.9 | 9.9 | 0.0 | 0.0 |
| 1 | 13.0 | 11.4 | 15.3 | 12.7 | **11.3** | **8.3** | **16.2** | **14.3** | 12.7 | 9.3 |
| 5 | 14.3 | 12.2 | 32.8 | 16.5 | 42.7 | 23.6 | 35.6 | 19.6 | 29.5 | 14.4 |
| 10 | 17.2 | 13.9 | 44.2 | 24.9 | 65.0 | 46.0 | 53.7 | 32.4 | 49.4 | 24.5 |
| 20 | 20.6 | 15.8 | 56.4 | 29.8 | 83.3 | 58.1 | 61.7 | 41.5 | 64.7 | 39.2 |
| 50 | 41.6 | 17.5 | 71.1 | 46.9 | 87.5 | 69.3 | 78.2 | 54.7 | 73.7 | 65.0 |
| 100 | 46.7 | 30.7 | 79.3 | 53.3 | 87.9 | 79.4 | 83.6 | 61.0 | 85.6 | 78.6 |

Table 5: Scene Coverage statistics for Training, SCMM, and Random Renderings. Location-only (**xyz**); Location-and-Orientation (**w/$\theta$**)

## 4.6 RETRIEVAL VISUALIZATION

In this section, we present the visualization results of our retrievals. Figure 5 showcases successful retrievals from various LERF scenes. As illustrated, the VLM used (XDecoder) accurately produces masks, and the associated visual embeddings closely align with the corresponding text embeddings, facilitating efficient matching. Importantly, these visual embeddings and masks are generated during the scene analysis phase, independent of any user text query. This feature ensures that the visual embeddings can be matched with any potential user query. We demonstrate additional failure cases in Figure 26.

## 5 CONCLUSION

We introduced the first end-to-end framework for retrieval of pre-trained neural graphics representations. By performing visual analysis on rendered RGB images, it is compatible with any form of pre-trained NGR. We propose novel techniques, Neural Graphics Noise Analysis and Smart Camera Movement Module, for detection of noise, navigation in NGR scenes, and selection of viewpoints for high-quality rendering. These high-quality renderings significantly improve retrieval accuracy. As a result, our solution achieves significantly higher retrieval accuracy than baselines such as LangSplat and LERF, even when training images are unavailable. Retri3D is thus general across NGRs, while being orders of magnitude more efficient in terms of speed and storage compared to existing techniques.

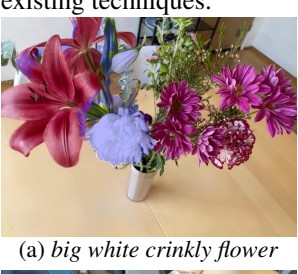

(a) *big white crinkly flower*

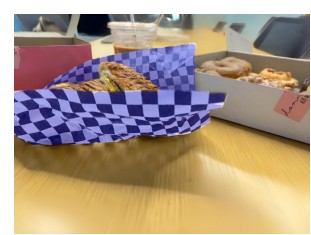

(b) *checkerboard pattern*

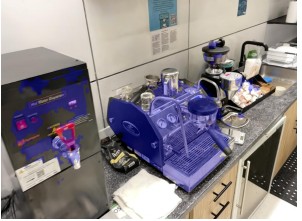

(c) *espresso machine*

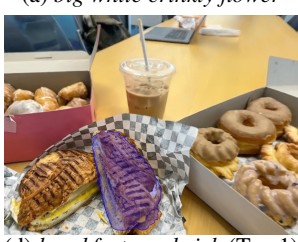

(d) *breakfast sandwich* (Top 1)

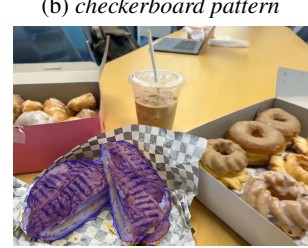

(e) *breakfast sandwich* (Top 2)

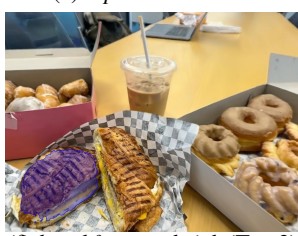

(f) *breakfast sandwich* (Top 3)

Figure 5: Successful Retrievals from LERF dataset with object label queries.

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

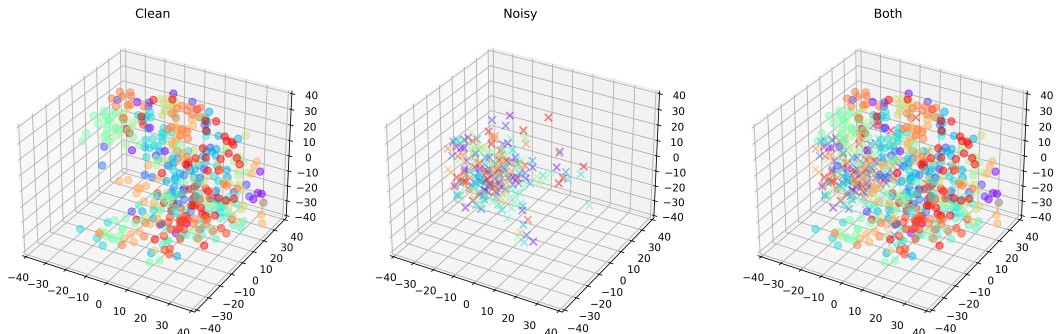

Figure 6: 3D t-SNE for clean pixel feature (O) and noisy pixel feature (X) for XDecoder's Feature Pyramid Network (Level 1, Activation Resolution: $512 \times 64 \times 88$)

# APPENDIX

## A    ADDITIONAL METHOD DETAILS

### A.1    VISUAL EMBEDDING GENERATION

To enable zero-shot performance on unseen scenes, our system is compatible with pre-trained VLMs without fine-tuning. This design choice contrasts with other NGR semantic analysis methods, which have high training and engineering overhead for each NGR format. Given a visual encoder of a VLM $\Phi$ parameterized by $\phi$, we generate the visual embeddings $\boldsymbol{E_v}$:

$$\boldsymbol{E}_v = \Phi(I; \phi) \in \mathbb{R}^{L \times d} \tag{8}$$

where $I \in \mathbb{R}^{3 \times H \times W}$ is the rendered RGB image, and $\boldsymbol{E_v}$ are $L$ visual embeddings of dimension $d$.

Different visual encoders produce varying numbers of embeddings per image. For example, CLIP generates a single embedding ($L = 1$) for the entire image (Cherti et al., 2022). More recent works, such as XDecoder produce multiple embeddings:

$$(\boldsymbol{E}_{whole}, \boldsymbol{E}_{seg}, \boldsymbol{M}_{seg}) = \Phi_x(I; \phi_x) \tag{9}$$

where $\Phi_x$ is a pre-trained XDecoder visual encoder parameterized by $\phi_x$. $\boldsymbol{E}_{whole} \in \mathbb{R}^{1 \times d}$ represents the whole image embedding, similar to a CLIP's embedding for the entire image. XDecoder also generates $K$ image segment embeddings ($\boldsymbol{E}_{seg} \in \mathbb{R}^{K \times d}$) with masks $\boldsymbol{M}_{seg} \in \mathbb{R}^{K \times H \times W}$ indicating the correspondence between the embedding and segments of the image. See Figure 5 for a visualization of these segments and the associated concepts.

In our system, we do not utilize all image segment embeddings from XDecoder, as some masks may not have a corresponding segment in the image. Specifically, we ignore image embeddings with a corresponding mask of less than 10 pixels as they usually don't correlate with any concepts in the image.

Our dataset then stores all the image embeddings generated via VLMs given all rendered images as inputs. During lookup, image embedding (and the associated scene) is retrieved based on the highest cosine similarity with the text embedding, generated from the user query $q$ via the VLM's text encoder $\Psi$, parameterized by $\psi$. Given scenes indexed by $n \in N$, rendered images of $m \in M$ per scene, the top scene/image/embedding is given by:

$$\arg\max_{n,m,l} \text{cossim}(e_v^{n,m,l}, e_t) \tag{10}$$

where $e_v^{n,m,l} \in \mathbb{R}^d$ is the $l$th image (segment) embedding for the $m$th image of the $n$th scene, and $e_t = \Psi(q; \psi) \in \mathbb{R}^d$ is the text embedding based on user query $q$.

Our system flexibly supports any VLM that employs image-text contrastive pre-training and is capable of independently generating visual and text embeddings with $\Phi$ and $\Psi$. This adaptability allows the system to benefit not only from advancements in pre-trained VLMs but also from fine-tuning on domain-specific visual or textual concepts when necessary. In this work, however, we test without fine-tuning to highlight the zero-shot performance while ensuring compatibility across any NGR format capable of rendering RGB images.

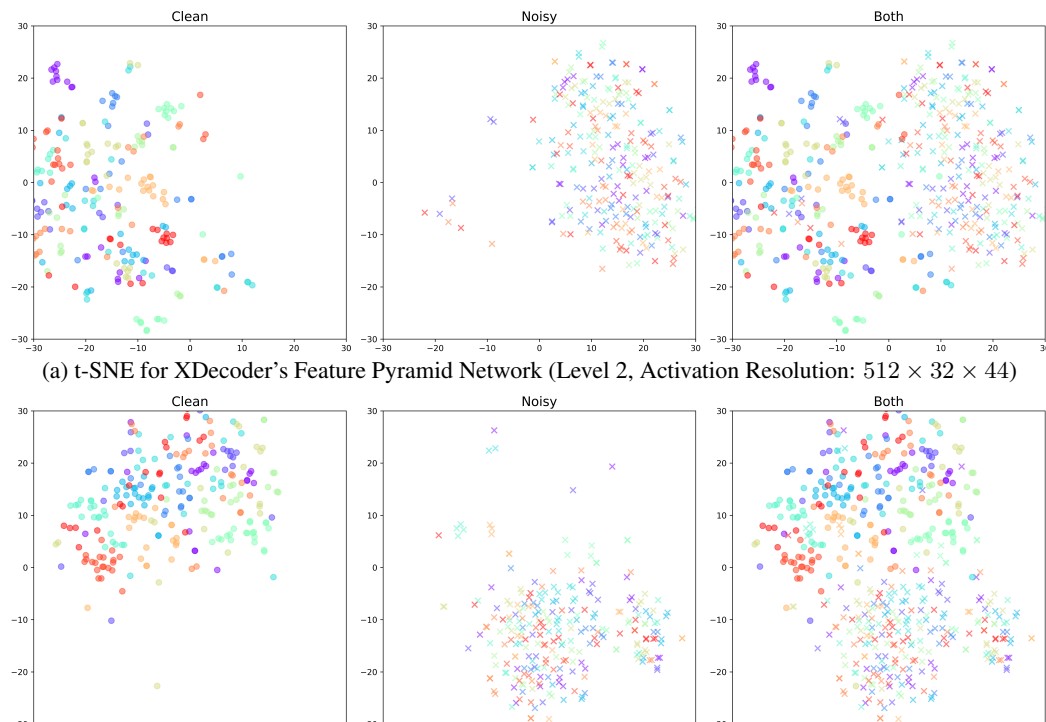

(a) t-SNE for XDecoder's Feature Pyramid Network (Level 2, Activation Resolution: $512 \times 32 \times 44$)

(b) t-SNE for XDecoder's Feature Pyramid Network (Level 3, Activation Resolution: $512 \times 16 \times 22$)

Figure 7: t-SNE visualization of clean pixel features (O) and noisy pixel features (X) for different levels of XDecoder's Feature Pyramid Network.

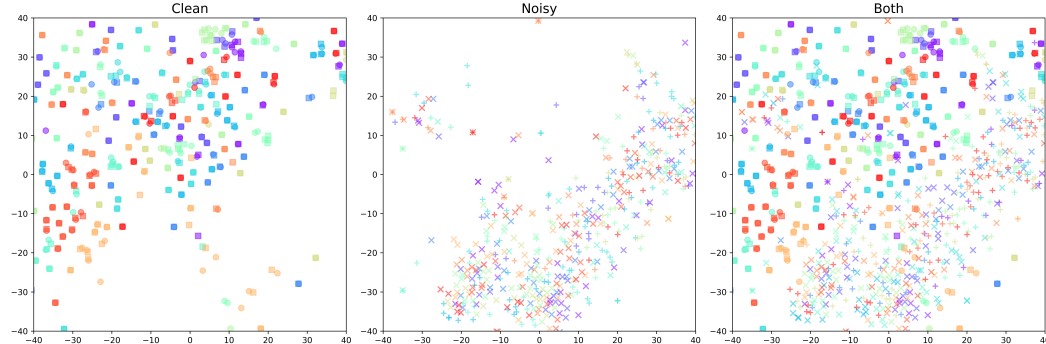

Figure 8: t-SNE for Nerfacto clean/noisy pixel feature (O/X) and Splatfacto clean/noisy pixel feature (□/+)

## A.2 NGR NOISE ANALYSIS VIA PRE-TRAINED VLMS

We demonstrate that the ability to generate well-separated features of clean pixels and noisy pixels is not a property specific to a certain layer of the XDecoder VLM, but applies to different layers of different VLMs. In addition, we also demonstrate that the noise features from NeRF (Nerfacto) and 3DGS (Splatfacto) are very close with each other, well-separated from the clean pixel features from the respective NGRs.

The t-SNE visualizations in Figure 2 and the rest in this section are generated via sampling activation features from the specified VLM layer given clean renderings and noise renderings. The clean renderings are generated via sampling 10 images (linspace) from the training image sequence of each scene; linspace is selected to ensure best diversification of the training clean images. The noise renderings are generated with 10 renderings at random viewpoints for each scene. We sample 2 activations per rendering randomly to generate the t-SNE plot. We use all 13 LERF scenes for the renderings. We select a small number of images and activations to avoid clustering and provide better visualization.

For calculation of the actual multivariate Gaussian distribution representing the noise as specified in Eq 1, we use 50 renderings from random viewpoints for each scene, and 50 activations from each of the renderings. The random viewpoints' camera origins are sampled uniformly from the center $1 \times 1 \times 1$ box of the scene. The view direction components $x$, $y$, and $z$ are sampled uniformly from [-1, 1], and the view direction vector is normalized. Note that we do not require the clean training images for calculating the noise Multivariate Gaussian distribution. XDecoder's image backbone ends with a Feature Pyramid Network. We use XDecoder's (*variant: X-Decoder-oq201*) Feature Pyramid Network's highest resolution level activation (as viewed in Figure 2 and 6) for noise multivariate Gaussian distribution training unless otherwise specified.

First, we provide a 3D tSNE of the clean and noisy features in Figure 6. The randomly sampled features are the same as the one demonstrated in the 2D t-SNE plot in Figure 2. We visualize them in 3D here to demonstrate that the clean features occupy a much larger volume. This shows that, in high dimensions, the noise features are much more tightly clustered than the clean features.

In Figure 7, we demonstrate the t-SNE plot for activations from other layers of the XDecoder visual encoder. Despite the lower resolution of the activation map for these feature pyramid network levels, the noise features and clean features are still well-separated.

In Figure 8, we demonstrate the t-SNE plot of pixel features from Nerfacto renderings and Splatfacto renderings. As seen in the plots, we observe that the noisy pixel features from the different NGRs are close to each other, separated from the clean pixel features from their respective NGRs. This means that using one type of NGR to train a multivariate Gaussian distribution to model the noisy regions from another NGR is possible. We will demonstrate the impact on retrieval accuracy with this cross-NGR noise Gaussian model training in Table 9.

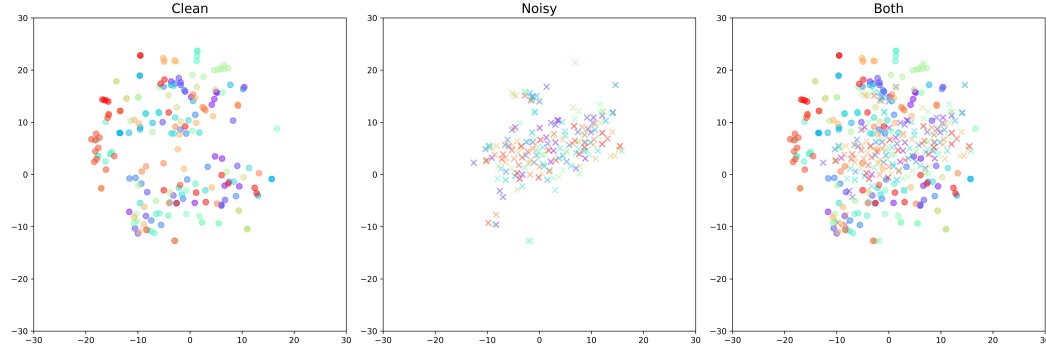

Figure 9: t-SNE for clean pixel feature (O) and noisy pixel feature (X) for the open-clip model ViT-L-14-336 (second to last transformer layer, Activation Resolution: $1024 \times 24 \times 24$)

Lastly, we visualize the t-SNE embeddings from the OpenCLIP model (*variant: ViT-L-14-336*) in Figure 9, which demonstrates a similar degree of separation between clean and noisy features. ViT-L-14-336 divides the image into $14 \times 14$ patches (tokens), and consists of 24 transformer layers. We utilize the activations from the penultimate (23rd) transformer layer to calculate the Gaussian distribution for noise. The final transformer layer is excluded as its patch-wise outputs are not leveraged during the contrastive training process, and thus lack sufficient semantic information. FLIP made a similar observation, and suggested performing token-wise contrastive learning to ensure the final layer's activations carry meaningful semantic content (**?**). However, we found this alternative training scheme to be unnecessary for distinguishing between noise and clean features. The activations from the penultimate layer of a standard contrastively trained vision transformer are sufficient for this task.

In summary, we observed that the noise features can be separated from clean features given different kinds of VLMs, as well as different kinds of NGRs.

### A.3 SMART CAMERA MOVEMENT IN 3D

In Section 3.4, we discussed how to rotate the camera to a viewpoint containing cleaner pixels given a single rendering. Here, we demonstrate that by rendering a set of images from viewpoints that vary in depth (w.r.t. the center image in its local coordinate), we allow the camera to change viewpoints for both translation and rotation, in order to navigate in 3D.

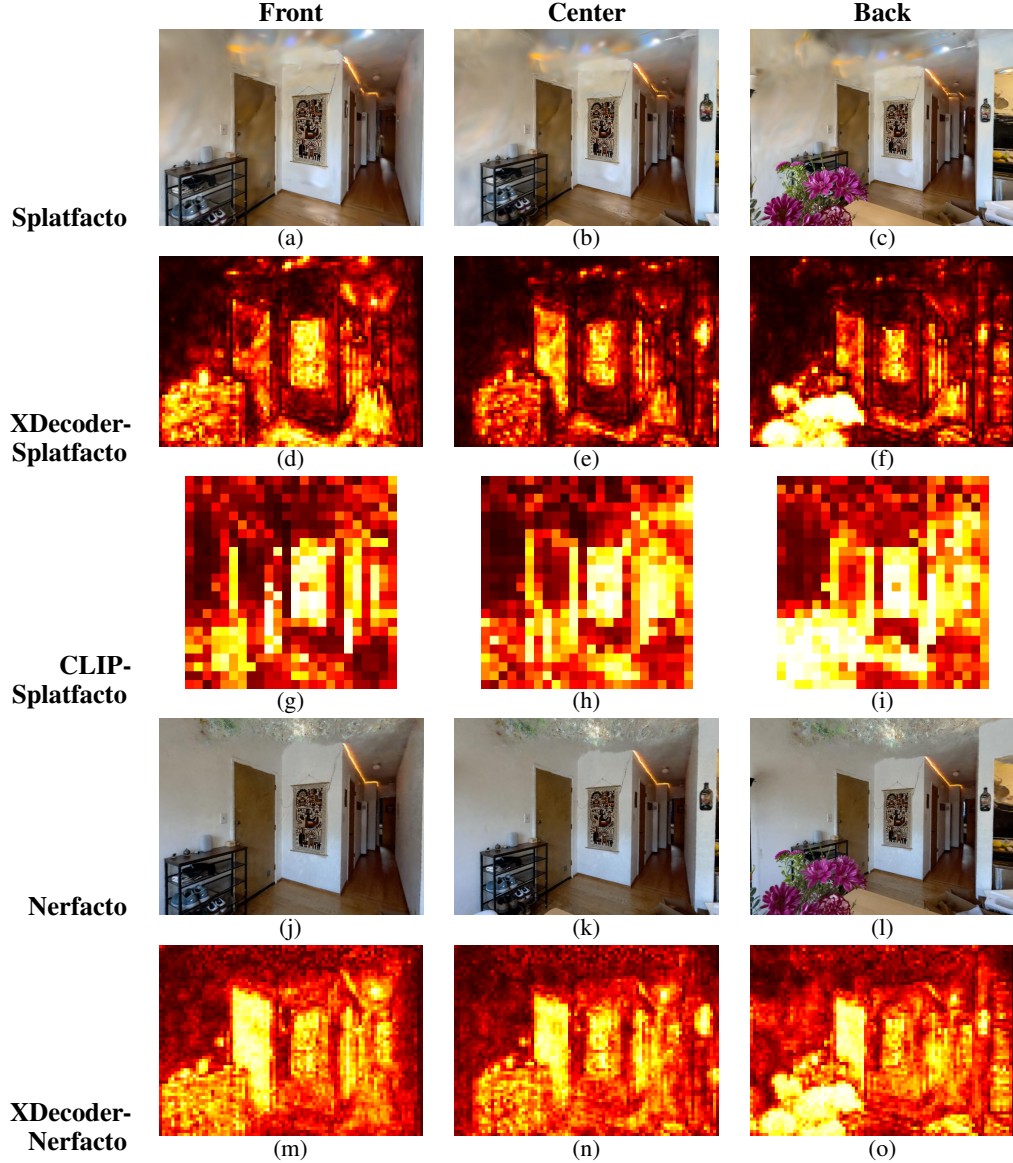

Figure 10: Splatfacto and Nerfacto Rendering with Varying Depth in Camera's Coordinate; And the Corresponding Clean Score Map from Different VLMs

Figure 10 visualizes a set of three images with identical rotation matrices but translated along the z-axis (depth) in the camera's local coordinates. Noise analysis is performed on all three images, and the optimal next viewpoint is selected based on the noise analysis result, aiming for the highest estimated clean score.

The result of this algorithm is a method for guiding the camera to move incrementally toward cleaner views of the scene, starting from a random initialization point. While SCMM with depth variation could be integrated into our retrieval system to produce higher-quality renderings and improve retrieval accuracy, we opt not to use this approach. The depth variation requires rendering a set of images instead of just one, and the associated computational cost is better allocated to selecting more initialization points to achieve greater scene coverage. However, our SCMM with depth demonstrates how navigation can be achieved using only a pre-trained NGR and VLM, which could benefit other works that interact with NGRs such as robot navigation. We provide an example of SCMM recovering from an extremely occluded viewpoint below.

In Figure 11, we show an example where the "center" current viewpoint is under extreme occlusion by floater artifacts. The "back" viewpoint provides some hints regarding where should the camera viewpoint point to for the next step. The SCMM algorithm leverages the analyzed noise to calculate

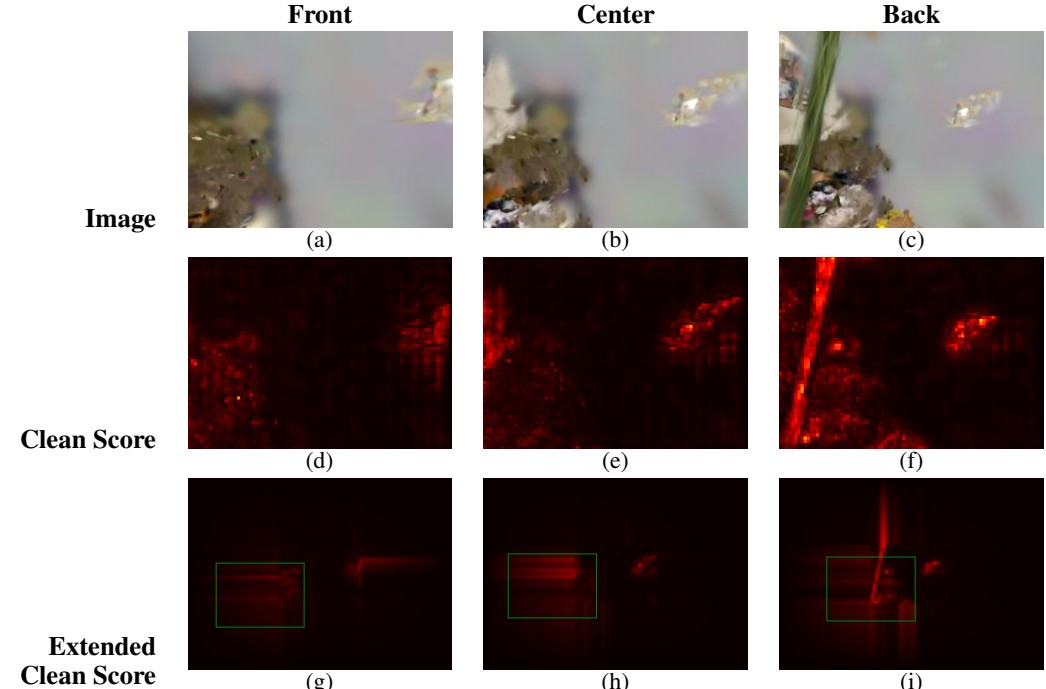

Figure 11: Visualization of Images, Clean Scores, and Extended Clean Scores for LERF Dozer Scene Rendering Set (Front, Center, Back).

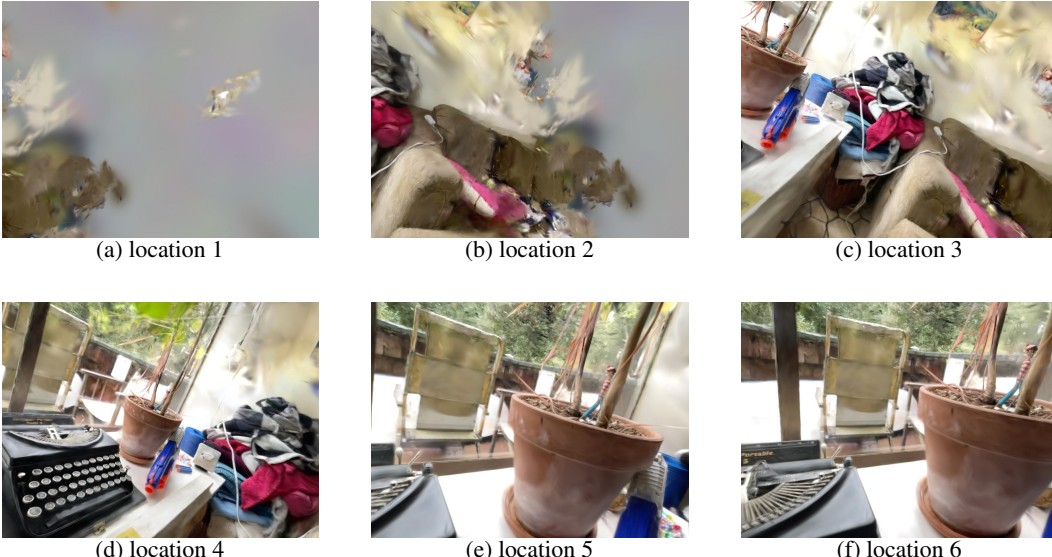

Figure 12: Dozer Scene Smart Camera Movement Recovering from Extremely Noisy Viewpoints via Movement in Depth

the extended clean score map for translation and rotation. Figure 12 shows how SCMM recovers from the poor initialization seen in Figure 11, iteratively moving the camera toward cleaner views. While our algorithm cannot guarantee successful recovery in every case, it often requires only a small hint, such as in this example, to complete the recovery process.

### A.4 EFFICIENT NOISE ANALYSIS AND SMART CAMERA MOVEMENT VIA LOW-RESOLUTION RENDERING

In order to render high-resolution clean images efficiently, we first render low-resolution images to perform noise analysis. We visualize this process in Figure 13. Given the noise analysis result, we move the camera to select viewpoints for cleaner images. From all the low-resolution images, we

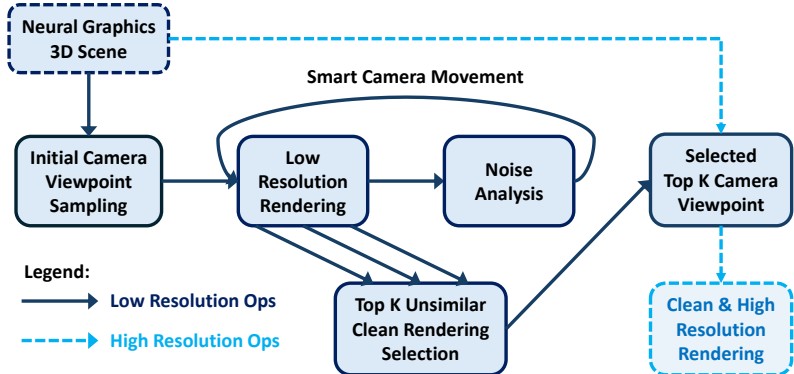

Figure 13: Process for Rendering of Clean and High Resolution Image from Noise Analysis and Smart Camera Movement with Low Resolution Renderings

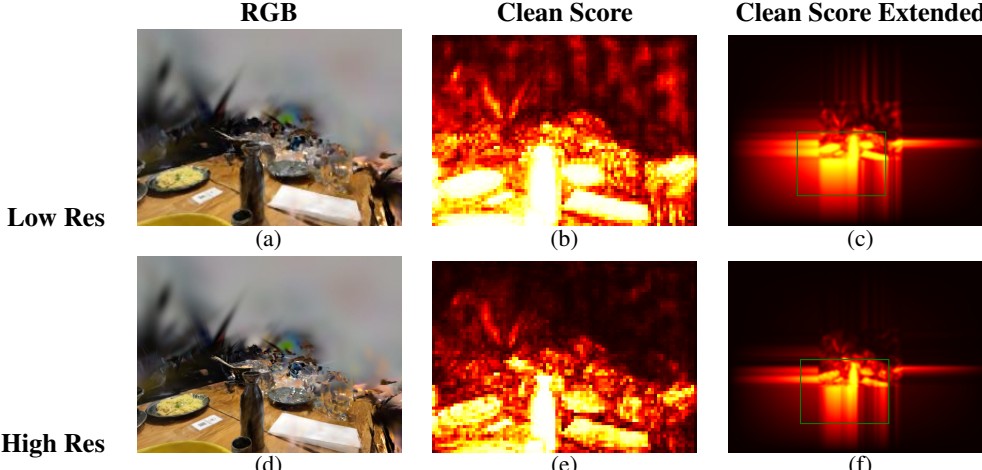

Figure 14: Low and High Resolution Rendering and Noise Analysis Comparison for Ramen Scene.

select the top-k not-similar images (image embeddings with cosine similarity less than 0.9) to render in high resolution. Note that most of the steps can be completed in the low-resolution space.

We visualize the noise analysis result for the same viewpoint, rendered in high resolution and low resolution (1/16 of the high-resolution in pixel count) for comparison in Figure 14. We use the same multivariate Gaussian distribution (trained through activations from high-resolution renderings). We note that despite a much lower resolution, the predicted noise remains mostly the same. The different predicted noises can lead to differences in terms of the exact camera movement, but both move the camera towards cleaner regions.

Such low-resolution rendering allows us to efficiently navigate within the NGR scene, adding only a few seconds overhead to the final high-resolution rendering and visual embedding generation. We provide quantitative results on the speed in Appendix B.8.

### A.4.1 CAMERA INITIALIZATION AND CONVERGENCE

We initialize the camera in SCMM by randomly sampling the camera origin components $x$, $y$, and $z$ independently from a Normal distribution with a standard deviation of 0.5. The view direction components $x$, $y$, and $z$ are sampled uniformly from [-1, 1], and the view direction vector is normalized. The same sampling method is applied to random viewpoint rendering. The Normal distribution sufficiently distributes the initial cameras, with a bias for more center views which is more likely to have partially noise-free views. Combined with uniform sampling of the viewing direction, this approach is more likely to provide different angles of the objects in the center while also capturing objects in the periphery. This approach provides good coverage of the scenes, while enabling the cameras to navigate to more noise-free views.

Our SCMM (rotation only, as used in the retrieval process described in the paper) is capable of recovering from poor initializations if the camera can be rotated to a better viewing direction without

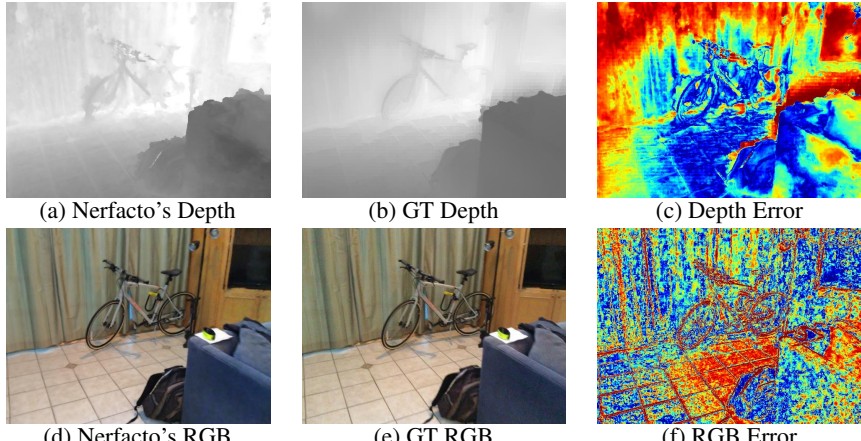

(a) Nerfacto's Depth      (b) GT Depth      (c) Depth Error

(d) Nerfacto's RGB      (e) GT RGB      (f) RGB Error

Figure 15: Depth and RGB Error for Image 1 of ScanNet Scene 0001 (Red is higher error; Blue is lower error.)

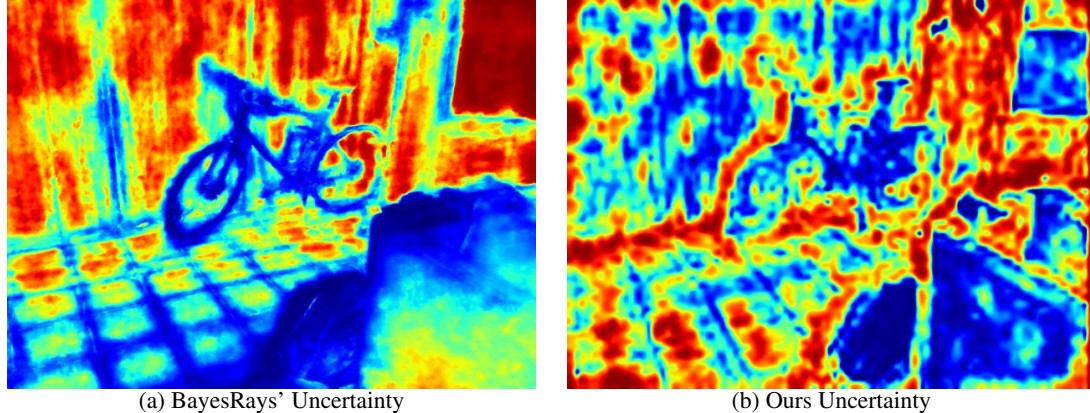

(a) BayesRays' Uncertainty      (b) Ours Uncertainty

Figure 16: Estimated Uncertainty for Bayes' Rays and Ours. (Red is higher uncertainty; Blue is lower uncertainty.)

moving. It converges to a local optimum as long as the estimation of the cleanliness level beyond the current field of view is reasonable, which we found to be almost always the case in practice. To recover from poor initializations where translation is required to achieve cleaner views (e.g., when the camera origin is inside artifact clouds or objects), SCMM with rotation and translation must be used. However, we do not use SCMM with rotation and translation for the retrieval accuracy measurements, as we found that using rotation only results in shorter scene analysis time while maintaining the same level of retrieval accuracy.

Should the ability to recover from poor initializations become critical, SCMM with rotation and translation can be employed. It is important to note that both versions of SCMM may still result in individual bad renderings. Therefore, sampling multiple renderings and rejecting bad ones is necessary to improve the collective visual semantics quality for the scene.

## A.5 COMPARISON OF NEURAL GRAPHICS NOISE ANALYSIS WITH NERF UNCERTAINTY ESTIMATION

Our predicted noise level through Neural Graphics Noise Analysis can be interpreted as a measurement of the uncertainty within the trained NGR scene. To evaluate this, we compare our noise (uncertainty)

| Scene # | 001 | 079 | 158 | 316 |
|---|---|---|---|---|
| Depth - Bayes' | **0.28** | **0.35** | **0.20** | **0.29** |
| Depth - Ours | 0.42 | 0.40 | 0.34 | 0.49 |
| RGB - Bayes' | 0.38 | 0.36 | 0.42 | 0.26 |
| RGB - Ours | **0.18** | **0.23** | **0.21** | **0.25** |

Table 6: AUSE for Bayes' Rays and Ours (Measured against Depth and RGB Error), Lower is Better

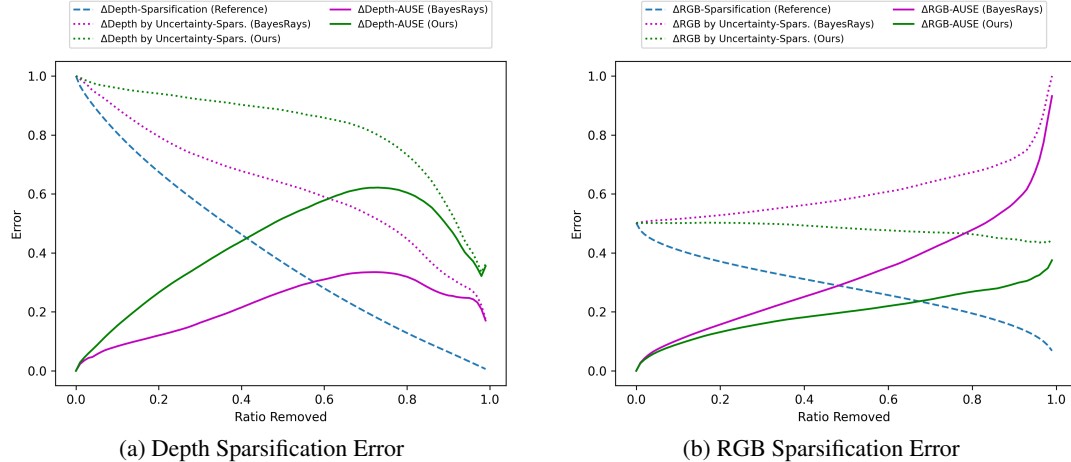

(a) Depth Sparsification Error  (b) RGB Sparsification Error

Figure 17: Sparsification Error for Bayes' Rays (Purple) and Ours (Green), Measured Against Depth and RGB Error. (Solid line represents AUSE, Lower is Better)

prediction with Bayes' Rays, the state-of-the-art method for uncertainty prediction in NeRF (Goli et al., 2023).

We note that other uncertainty estimation methods exist. For example, several works modify the NGR neural network to output uncertainty using Bayesian Neural Networks (Shen et al., 2021a; Pan et al., 2022; Shen et al., 2022). Niko et al. proposed an ensemble of NeRFs to estimate uncertainty, which is computationally expensive (Sünderhauf et al., 2022). FisherRF uses the Fisher information matrix between learned parameters of NGRs, but this is only applicable to models like 3DGS and Plenoxels with sparse, uncorrelated parameters (Jiang et al., 2023). Bayes' Rays perform post-hoc training on NeRFs, requiring both RGB and density information from the NeRF MLPs (Goli et al., 2023). Since Bayes' Rays achieved the state-of-the-art performance, we choose to compare our solution with Bayes' Rays.

Assessing the quality of estimated uncertainty is challenging. Bayes' Rays proposed using the Area Under Sparsification Error (AUSE), measured against depth. The idea is that the pixels are sparsified (gradually removed) in two ways: first by removing pixels based on depth error (highest depth uncertainty pixels are removed first), and second by removing pixels based on predicted uncertainty (highest uncertainty pixels are removed first). Ideally, higher uncertainty predictions should correspond to greater depth error, resulting in both sparsification processes retaining similar errors. The gap of remaining errors between these two sparsification processes constitutes the AUSE-Depth error. As shown in Figure 17(a), Bayes' Rays achieves a lower (better) AUSE-Depth than our method. This is expected since Bayes' Rays estimates spatial uncertainty via neural network perturbations within NeRF's MLP.

However, when we conduct the same AUSE measured against RGB error, our method achieves a lower (better) AUSE-RGB compared to Bayes' Rays (Figure 17(b)). This indicates that our approach more effectively estimates uncertainty that aligns with RGB rendering errors, which is reasonable given that our uncertainty (noise) is derived from RGB renderings in the first place. We show the quantitative comparison for the 4 scenes provided in Bayes' Rays dataset in Table 6. On all four scenes, we perform better for AUSE-RGB while Bayes' Rays perform better for AUSE-Depth.

To generate our uncertainty predictions, we simply use the multivariate Gaussian distribution trained with LERF scene noise renderings. Using this Gaussian distribution, we estimate the noise level for the ScanNet scene's rendering from a Nerfacto model. Our multivariate Gaussian distribution did not use renderings from ScanNet scenes during training. Unlike Bayes' Rays, our method does not require access to or post-processing of the NGR model. Moreover, our approach is applicable to both 3DGS and NeRF, while Bayes' Rays is limited to NeRF-based solutions.

While our method outperforms Bayes' Rays in AUSE-RGB and Bayes' Rays excels in AUSE-Depth, we observe notable differences from all of the error by depth, error by RGB, and uncertainty predictions from both Bayes' Rays and our method. This highlights a need for further research into uncertainty estimation and the development of appropriate metrics to assess their effectiveness.

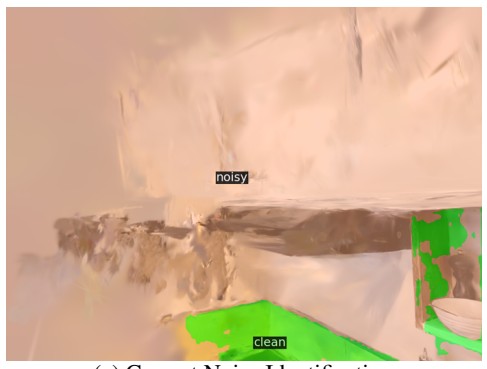 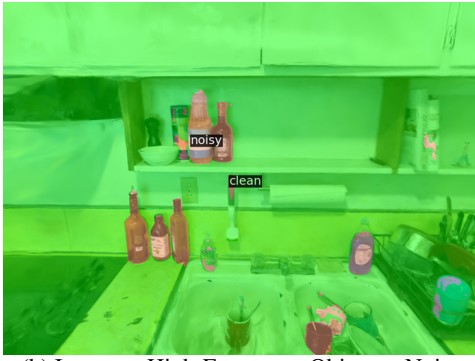

(a) Correct Noise Identification        (b) Incorrect High-Frequency Object as Noise

Figure 18: Noise Identification via Finetuning of XDecoder Segmentation Head

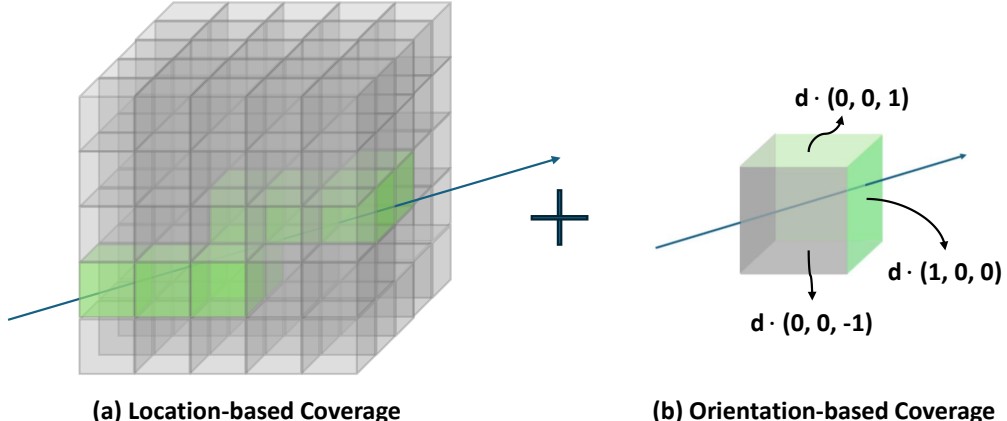

**(a) Location-based Coverage**        **(b) Orientation-based Coverage**

Figure 19: Coverage Map Visualization

### A.6 NOISE ANALYSIS THROUGH VLM CLASSIFICATION

In addition to modeling noise with multivariate Gaussian distribution, another viable approach is to fine-tune a segmentation model to segment the noise regions from clean regions. In summary, we find that a fine-tuned XDecoder often treats high-frequency objects as noise, as seen in Figure 18. This is particularly detrimental for scene retrieval as it will direct cameras away from interesting regions containing high-frequency objects.

To test the idea, we fine-tune an XDecoder model's segmentation head to classify between clean and noisy. Specifically, we use the same random viewpoint renderings that were used to fit the multivariate Gaussian distribution (50 from each of Splatfacto NGRs representing the 13 LERF scene), coupled with 50 training viewpoint renderings from each scene. We modify the last layer of the segmentation head so it only produces two classes, trained to predict all training viewpoint renderings as clean, and all random viewpoint renderings as noise. We unfreeze the segmentation head parameter and train it for 35 epochs with SGD (lr=1e-4, momentum=0.9, weight decay=1e-4). We also tried unfreezing the Feature Pyramid Network but obtained similar results.

A more ideal fine-tuning solution would involve using images with proper segmentation of noise and clean regions. However, we found this to be a very ambiguous task not easily carried out by manual labor. As we have discussed in Appendix A.5, existing uncertainty estimation methods do not generate high-quality segmentations (particularly when evaluated using RGB error) automatically either.

### A.7 SCENE COVERAGE CALCULATION

We evaluate scene coverage for different rendering methods within the spatially contracted space of Nerfacto. Nerfacto uses a spatial contraction technique modified from MipNeRF-360 (Barron et al., 2021), which warps infinite distance into a box of size $4 \times 4 \times 4$. We divide the box into a grid of

$64 \times 64 \times 64$ voxels. We did not evaluate the coverage with Splatfacto as it is difficult to measure such a statistics in an infinite space, but we expect similar conclusions to hold for both NGRs.

To update the coverage statistics within this grid, we perform ray traversal for each rendering. Rays are traced from the near plane of the rendering to the estimated depth predicted by Nerfacto, marking any voxel they traverse through as covered. For the location+orientation statistics, we compute the dot product between the ray and the normal vector of the voxel's face. We visualize this in Figure 19. If the dot product is positive, the orientation is recorded as observed. Although this is a relatively relaxed criterion, it is consistently applied across all rendering types.

Before rendering, we also sample the density at the center of each voxel. If a voxel's density exceeds 0.5, we mark the entire voxel, along with all its orientations, as covered.

For the %Train measurement, we track all voxels (and faces) covered by rays from the training renderings, without pre-filling the grid with coverage information based on density. We then calculate the percentage of voxels (and faces) covered by the training rays that are also covered by other rendering types.

## B  ADDITIONAL RESULTS

### B.1  EXPERIMENT SETTING

**Dataset**  We utilize the LERF and ScanNet++ datasets, comprising 13 scenes and 280 scenes respectively (Kerr et al., 2023; Yeshwanth et al., 2023). Only ScanNet++ scenes with object labels are included.

**Text Query**  We employ the original object labels from LERF and ScanNet++ as text queries. Additionally, we generate image captions from the training images using the LLaVA (v1.5-13b) model with the prompt: *Describe the image in detail in one sentence*. It is important to note that the VLM used in our retrieval (XDecoder) never observes the exact training images at any point. Furthermore, the LLaVA model is developed independently of XDecoder (Liu et al., 2023b; Zou et al., 2022).

**Neural Graphics Representations**  We use the nerfstudio implementations of 3D Gaussian Splatting and NeRF, specifically Splatfacto and Nerfacto (Tancik et al., 2023). All models are trained using the default configuration for 30,000 epochs. For Nerfacto, we also enable the *"use-gradient-scaling"* option to scale the gradient near the camera, reducing artifacts and creating a stronger baseline for rendering from random poses (Philip & Deschaintre, 2023). LERF employs the same Nerfacto as its RGB rendering component. LangSplat uses the original 3DGS implementation, while the nerfstudio Splatfacto aims to replicate 3DGS but may have minor implementation differences (Tancik et al., 2023; Qin et al., 2023; Kerbl et al., 2023).

**Metrics**  A retrieved scene is considered correct if it matches the user query. For object labels, this means the scene contains the queried object. For LLaVA image captions, this means the scene was used to generate the caption. Note that LERF has minimal label overlap between scenes, whereas ScanNet++ has significant overlap. We measure precision P@k=1,5,10, indicating whether the top-k retrieved scenes include the correct scene.

**Experimental Platform**  Experiments are conducted on a desktop with an Intel i7-13700K CPU, Nvidia RTX 4090 GPU, and 64GB of RAM. We use PyTorch 2.1.2 with CUDA 12.0 on Ubuntu 22.04 LTS.

### B.2  DIFFERENT VLM

In addition to XDecoder, we also evaluate our system's performance when other VLMs are used. In particular, we select open-clip (Cherti et al., 2022) to generate the visual embeddings and text embeddings. Compared to XDecoder, open-clip does not perform segmentation or generate multiple embeddings (corresponding to different segments) per image. It involves a simple process of generating one image embedding per image. We use this image embedding that describes the entire image for retrieval in our system.

In Table 7, we compare the retrieval performance of our system when the XDecoder or CLIP model is used. Regardless of the viewpoint choice, our system achieves higher performance when XDecoder is used. This is understandable as XDecoder is a more recent solution, and it also captures sub-image contents in addition to the whole image embedding. We note that the Random viewpoint rendering is particularly difficult for CLIP to generate any meaningful embedding. Even with 100 images being

| # | XDecoder | | | OpenCLIP | | |
| | Viewpoint | | | Viewpoint | | |
| Img. | Training | SCMM | Random | Training | SCMM | Random |
|---|---|---|---|---|---|---|
| 1 | **57.89** | 30.08 (-27.81) | 17.29 (-40.60) | **41.35** | 20.59 (-20.76) | 11.63 (-29.72) |
| 5 | **68.42** | 64.66 (-3.76) | 18.80 (-49.62) | **60.90** | 46.83 (-14.07) | 13.45 (-47.45) |
| 10 | **78.95** | 67.67 (-11.28) | 24.81 (-54.14) | **61.65** | 47.24 (-14.41) | 16.43 (-45.22) |
| 20 | **83.46** | 73.68 (-9.78) | 34.59 (-48.87) | **63.16** | 50.47 (-12.75) | 18.24 (-44.92) |
| 50 | **84.21** | 77.69 (-6.52) | 49.62 (-34.59) | **64.62** | 53.79 (-10.83) | 22.49 (-42.13) |
| 100 | **84.95** | 80.02 (-4.93) | 57.89 (-27.06) | **65.29** | 57.68 (-7.61) | 24.93 (-40.36) |

Table 7: Retrieval Accuracy (P@1) for Splatfacto Model on LERF dataset. Comparison between XDecoder and OpenCLIP.

| # | Viewpoint | | |
| Img. | Training | SCMM | Random |
|---|---|---|---|
| 1 | 57.89 / 81.95 / 97.74 | 30.08 / 80.28 / 95.96 | 17.29 / 54.14 / 83.46 |
| 5 | 68.42 / 93.98 / 98.50 | 64.66 / 91.69 / 97.09 | 18.80 / 48.87 / 85.71 |
| 10 | 78.95 / 96.24 / 99.25 | 67.67 / 95.69 / 99.25 | 24.81 / 58.65 / 89.47 |
| 20 | 83.46 / 98.50 / 100.0 | 73.68 / 97.82 / 99.68 | 34.59 / 69.17 / 93.23 |
| 50 | 84.21 / 99.25 / 100.0 | 77.69/ 98.23 / 100.0 | 49.62 / 80.45 / 92.48 |
| 100 | 84.95 / 98.50 / 100.0 | 80.02/ 98.61 / 100.0 | 57.89 / 87.22 / 99.25 |

Table 8: Retrieval Accuracy (P@1/5/10) for Splatfacto on LERF Dataset with LERF Object Labels

rendered per scene, it still has a significant gap with the Training viewpoint rendering. This is due to the noisy component in the images severely hindering CLIP's ability to generate quality image embedding. For XDecoder, even when large sections of the image are noise, it can still generate some image embeddings highly associated with the clean regions in an image.

The CLIP model's performance demonstrates the importance of selecting noise-free high quality view when performing 3D NGR analysis with VLMs.

### B.3 SPLATFACTO P@K RETRIEVAL ACCURACY

In Table 8, we report top-k retrieval accuracy for the Splatfacto model. As shown the in table, the retrieval accuracy for k equals to 5 and 10 increases as the top 1 retrieval accuracy increases.

### B.4 NERFACTO OBJECT LABEL AND LLAVA CAPTION RETRIEVAL

In Table 9, we show the result for Nerfacto model on the LERF dataset. Compared to Splatfacto's result in Table 1, we see comparable results. The object label retrieval accuracy slightly decreases but the LLaVA caption accuracy slightly increases. We believe this is a result of the Nerfacto model being worse at capturing small details of the objects but still having good overall image rendering quality.

In addition, we also demonstrate the retrieval accuracy when different sources are used for training the multivariate Gaussian distribution for noise. Specifically, we use either Gaussian distributions trained from Nerfacto noise renderings or Splatfacto noise renderings. We found the training from the same source as the model (Nerfacto) to work slightly better, but the different in accuracy is minimal.

### B.5 MULTIPLE QUERIES RETRIEVAL

As the scenes are complex and may contain similar objects, users may want to retrieve a scene that satisfies several descriptions or contains several objects. We perform another experiment allowing the users to specify multiple queries at a time.

For evaluation purposes, we use combinations of 1, 2, or 5 object labels from LERF. To combine the queries, we calculate the product of the exponential of the maximum cosine similarity for each scene with respect to each query ($q$). The top scene, indexed by $n$, is identified as follows:

$$n = \arg\max_n \prod_{q=1}^{Q} \left( exp(1 + \max_{l \in \{1,...,L\}} \text{cossim}(e_{n,l}, e_q)) \right)$$

where L is the total number of visual embeddings for each scene. A notable challenge arises when all embeddings are stored in a single database, requiring the calculation of cosine similarities for all stored embeddings to retrieve the top scene, which is highly inefficient. To address this, we can

| Noise Gaussian | Object Label | | | | LLaVA Caption | | | |
|---|---|---|---|---|---|---|---|---|
| | Training | SCMM | | Random | Training | SCMM | | Random |
| | | Nerfacto | Splatfacto | | | Nerfacto | Splatfacto | |
| 1 | 52.79 | 24.24 | 23.89 | 14.86 | 55.37 | 23.03 | 21.46 | 17.6 |
| 5 | 63.60 | 61.45 | 59.36 | 19.37 | 64.08 | 67.64 | 66.73 | 21.54 |
| 10 | 67.21 | 65.74 | 64.68 | 19.82 | 69.30 | 71.52 | 70.25 | 29.64 |
| 20 | 76.76 | 65.52 | 63.60 | 34.23 | 71.42 | 73.84 | 72.01 | 40.76 |
| 50 | 75.86 | 72.43 | 71.92 | 40.09 | 75.09 | 75.18 | 74.55 | 49.73 |
| 100 | 78.50 | 72.23 | 72.50 | 44.59 | 75.51 | 75.63 | 74.15 | 55.86 |

Table 9: Retrieval Accuracy (P@1) for Nerfacto Model on LERF dataset; Noise Gaussian indicates the Source of Training for the Noise Multivariate Gaussian Distribution

| # Img. | Training | SCMM | Random |
|---|---|---|---|
| 1 | 57.89 / 63.85 / 64.69 | 30.08 / 29.62 / 29.02 | 17.29 / 17.69 / 12.41 |
| 5 | 68.42 / 83.82 / 91.08 | 64.66 / 73.85 / 89.86 | 18.80 / 18.85 / 12.59 |
| 10 | 78.95 / 91.15 / 98.43 | 67.67 / 83.08 / 94.76 | 24.81 / 25.38 / 19.58 |
| 20 | 83.46 / 95.00 / 98.08 | 73.68 / 84.23 / 95.45 | 34.59 / 36.15 / 31.99 |
| 50 | 84.21 / 95.77 / 99.65 | 77.69 / 88.46 / 99.65 | 49.62 / 57.69 / 64.34 |
| 100 | 84.95 / 96.54 / 99.83 | 80.02 / 90.01 / 99.48 | 57.89 / 71.15 / 87.76 |

Table 10: Retrieval Accuracy (P@1) for Splatfacto on LERF dataset with Multiple Labels (1/2/5)

store a set of embeddings per scene, making the calculation scale linearly with the number of scenes instead of the number of visual embeddings. If multiple query retrieval is unnecessary, storing all visual embeddings in a single database remains an option.

Overall, accuracy significantly improves as the number of labels increases, achieving over 99% accuracy when 5 labels are used.

## B.6 MIXED NGR RETRIEVAL ACCURACY

In Table 11, we show the retrieval accuracy when a mixture of Splatfacto and Nerfacto models are stored in the database. In our system, rendering, and then performing analysis from different models is very easy. As shown in the table, our system achieves a high level of accuracy even when the stored NGRs are different.

## B.7 SCANNET++ RETRIEVAL ACCURACY COMPARED WITH LANGSPLAT

In Tables 12 and 13, we present a comparison between our method and LangSplat on the ScanNet++ dataset, using either object labels or LLaVA caption sentences as queries. To make the database for storing LangSplat features manageable, we only store 200 embeddings per image, randomly selected from all the embeddings. Our approach consistently achieves higher retrieval accuracy than LangSplat, particularly when training viewpoints are available. The performance gap is even more pronounced (almost double) when LLaVA captions are used, as LangSplat struggles to handle whole image descriptions effectively. Given that ScanNet++ contains highly similar scenes (offices, bedrooms, living rooms, etc.), we also adapt our multi-object-label approach (Appendix B.5), using it on multiple sentences. This multi-sentence is unique to our system, as our efficient query process leverages the small number of embeddings generated during scene analysis. The rapid querying capability allows us to perform multiple queries with additional sentences, leading to a more robust collective decision.

## B.8 VIEW SELECTION EFFICIENCY ANALYSIS

In Table 14, we perform a speed comparison between the low/high resolution rendering and noise analysis, as well as the total time for using and not using the hierarchical view selection process. As seen in the table, rendering 80 additional low-resolution images for each set of 20 high-resolution images adds less than 50% overhead. Since our algorithm is already very fast in analyzing a 3D scene, such overhead is relatively small, and the retrieval accuracy is much better compared to rendering more images from random viewpoints.

| # | Object Label | | | LLaVA Caption | | |
|---|---|---|---|---|---|---|
| | Viewpoint | | | Viewpoint | | |
| Img. | Training | SCMM | Random | Training | SCMM | Random |
| 1 | **54.28** | 28.46 | 14.92 | **49.43** | 33.24 | 14.24 |
| 5 | **65.68** | 62.84 | 15.84 | **67.04** | 59.26 | 24.67 |
| 10 | **72.24** | 63.12 | 22.42 | **69.00** | 62.85 | 26.85 |
| 20 | **79.49** | 75.43 | 32.43 | **71.42** | 64.63 | 31.46 |
| 50 | **80.02** | 77.43 | 46.74 | **72.80** | 65.43 | 33.75 |
| 100 | **80.67** | 79.62 | 54.34 | **74.63** | 71.86 | 45.64 |

Table 11: Retrieval Accuracy (P@1) for a 7-to-6 mix of Splatfacto Model and Nerfacto Model Representing the LERF scenes

| # | Object Label | | |
|---|---|---|---|
| | Ours | | LangSplat |
| | Viewpoint | | Viewpoint |
| Img. | Training | SCMM | Training |
| 10 | **41.62** | 39.63 | 38.19 |
| 20 | **50.06** | 47.34 | 43.68 |
| 50 | **58.23** | 54.39 | 51.52 |

Table 12: Retrieval Accuracy (P@1) for Splatfacto Model with Object Labels for ScanNet++ Scenes. Comparison between ours and LangSplat.

### B.9 LERF AND LANGSPLAT EXPLAINATIONS

In this section, we discuss the implementation of LERF and LangSplat and address the issues in terms of computation and storage when adapting these methods to a retrieval system.

**LERF** LERF embeds multi-scale image embeddings from open-clip into a "semantic radiance field" (Cherti et al., 2022). While the original RGB radiance field only outputs 4 values (RGB+density), the semantic field outputs features of size 512 as open-clip image features have size 512. To encode such large features effectively, significantly larger MLPs and hash-tables are used for the semantic field compared to the RGB field encoding. For LERF, overhead in computing and storage for both training and rendering stems from this excessively large semantic field.

**LangSplat** LangSplat uses the 3D Gaussians to encode the semantic features. They used SAM to segment the training images into different segments and used open-clip to encode each segment into an embedding to train a "Semantic 3D Gaussian" (Kirillov et al., 2023). However, instead of encoding features of size 512, they encode features of size 3. This is achieved via training a *per-scene* autoencoder that maps the feature of size 512 to a feature of size 3 during training. The "Semantic 3D Gaussian" is relatively small since it only encodes feature size 3. However, such a feature is not comparable between scenes as the autoencoders are trained on a per-scene basis. To perform retrieval across different scenes, we decode the features to size of 512 using the trained autoencoders, and store them in the database. In summary, for training, the LangSplat overhead comes for open-clip embedding generation for a large number of segments created by SAM and training the autoencoder. For rendering, the LangSplat overhead comes from converting pixel-wise size 3 feature to size 512 feature with the trained autodecoder.

### B.9.1 LERF AND LANGSPLAT SPEED AND STORAGE

For NGR Training, our methods uses the nerfstudio's Splatfacto and Nerfacto implementations as is. Our training of the RGB-only NGRs resulted in significantly simpler pipelines and lower overhead than LERF and LangSplat. Below, we will explain the LERF and LangSplat overhead at a high level, and would refer readers to their paper for more details (Kerr et al., 2023; Qin et al., 2023).

LERF's training overhead comes from training the field to output visual embeddings alongside the RGB values. The visual embedding's size (512) is significantly larger than the RGB's size (3). As a result, LERF chose to use a large hash-table feature grid to store and learn the visual embeddings. In addition, they used a multi-scale visual embedding, which included 30 levels of scales. If one renders all 30 levels of scales, the total embedding size for 1 image is 45GB (assuming a resolution of 994 x 738 as in the LERF data). In LERF, only one scale is selected during the interactive session for visualization purposes. However, such a selection depends on the text prompt. In a retrieval setting like ours, this is impossible. Therefore, to make implementation feasible, we chose 3 scales instead of 30, which still leads to a very large size but experiments could be conducted.

| | LLaVA Caption | | | | | |
|---|---|---|---|---|---|---|
| | Ours | | | | | LangSplat |
| # | Viewpoint | | | | | Viewpoint |
| Img. | Training 1 Sent | SCMM 1 Sent | SCMM 5 Sent | SCMM 10 Sent | SCMM 20 Sent | Training |
| 10 | 20.56 | 19.12 | 31.46 | 35.66 | **36.24** | 8.17 |
| 20 | 26.68 | 24.69 | 28.92 | 37.72 | **39.79** | 9.52 |
| 50 | 30.92 | 27.57 | 29.46 | 40.53 | **43.58** | 11.28 |

Table 13: Retrieval Accuracy (P@10) for Splatfacto Model with LLaVA Captions for ScanNet++ Scenes. We optionally use multiple sentences for retrieval. Comparison between ours and LangSplat.

| | 1x Low Res | 1x High Res | 20x High Res | 80x Low + 20 High |
|---|---|---|---|---|
| Rendering Time | 0.0029 | 0.036 | 0.72 | 0.952 |
| Noise Analysis Time | 0.005 | 0.005 | 0.31 | 0.5 |
| Total Time | 0.0079 | 0.041 | 1.03 | 1.45 |

Table 14: Rendering Technique Speed Comparison

To further complicate things, LERF uses negative prompts to increase the relevancy of the positive prompt. To calculate the relevancy in consideration of negative prompts, one need simultaneous access to the positive prompt embedding, LERF field encoded visual embedding and negative prompt embedding. From a retrieval perspective, this requires calculation between the positive query embedding with all visual embeddings (which is 4GB per image assuming 3 scales and 45GB per image assuming 30 scales). This is impossible to calculate directly. In our testing, we first select the top 10 images that have the highest cosine similarity match, and then calculated the relevancy score as specified in the LERF paper at 3 scales.

LangSplat's actual 3DGS process is relatively fast, after the main RGB 3DGS is trained, a secondary one that encodes the visual embeddings (in size 3) is trained on top of it. Its main overhead comes from the segmentation of the images (with SAM), and encoding each segmentation separately (CLIP), and training a per-scene autoencoder that maps the visual embedding from size 512 to 3. LangSplat is slow as its rendered visual embedding (size 3) need to be decoded to size 512 with the autoencoders cross-scene comparison. This is expensive as one visual embedding is rendered for every 4 pixels, and the autoencoders need to be run many times.

### B.10 LERF VISUALIZATIONS

As shown on the left of Figure 20, the cosine similarity exhibits almost no variation despite the query *coffee mug* should respond to the "coffee mug" in the top scene and "coffee cup" in the bottom scene. Variations in the cosine similarity only become visible in the middle column, where the min and max cosine similarity is used to create the color scale. On the right column, we demonstrate a relevancy map calculation based on the 3 multi-scale features we can store on disk for lookup (instead of the 30 scales originally used in the LERF paper). As shown in the right column, the *coffee mug* is incorrectly matched with the "coffee cup" in the incorrect scene. But most importantly, it is not showing relevance with the coffee mug in the correct scene.

In summary, adapting solutions like LERF for the retrieval process is non-trivial, leading to significant storage and speed overhead while reaching low accuracy.

## C ADDITIONAL VISUALIZATIONS

### C.1 ADDITIONAL SMART CAMERA MOVEMENT VISUALIZATION

In Figure 21, we visualize the smart camera movement allowing movement in depth for the camera origin.

In Figure 22, we visualize smart camera movement without movement in depth, which is what we use in the system for retrieval. As shown in the image, from a viewpoint looking at the ceiling, which is an uncovered area during training image sequence capture, our solution moves to a clear view of the scene in two steps.

In Figure 23, we show another challenging case where the initialization point is mostly random noise. Our Smart Camera Movement efficiently moves the viewpoint to a location where the person can be clearly viewed.

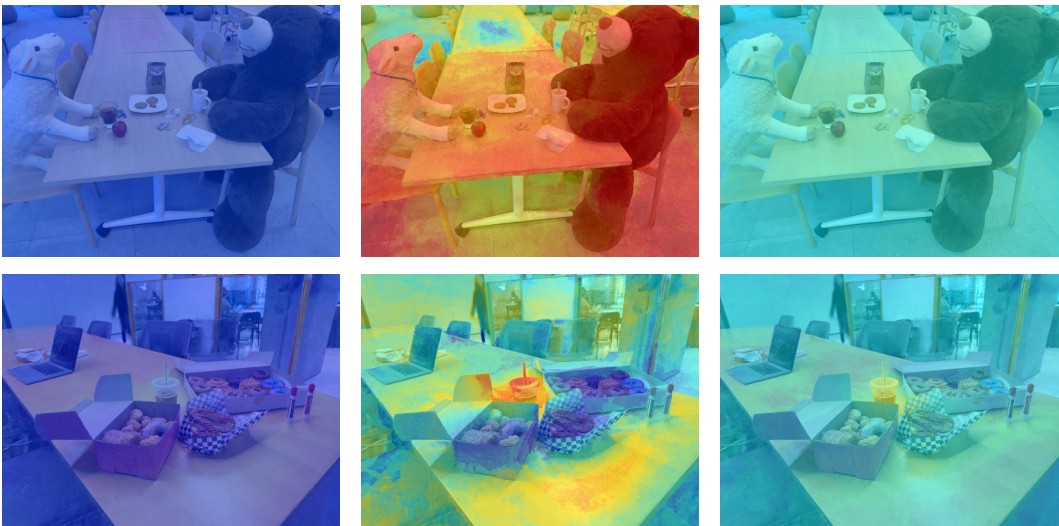

Figure 20: LERF cosine similarity and relevancy map given query *coffee mug*. Left: cosine similarity visualized in scale [-1, 1]. Middle: cosine similarity scaled [min cos-sim, max cos-sim] of the individual image. Right: relevancy map calculated with the assist of negative prompts.

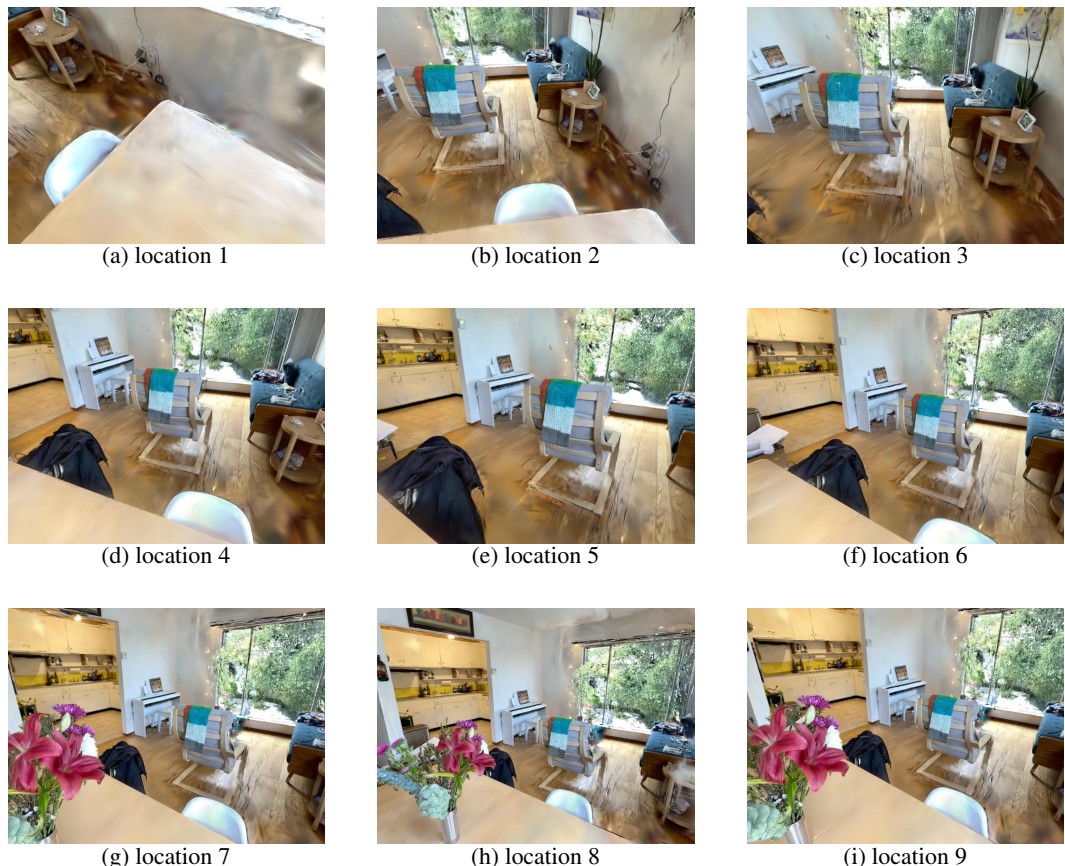

| (a) location 1 | (b) location 2 | (c) location 3 |
| (d) location 4 | (e) location 5 | (f) location 6 |
| (g) location 7 | (h) location 8 | (i) location 9 |

Figure 21: Camera Movement Allowing Changes in Depth

## C.2 SMART CAMERA MOVEMENT COMPARISON BETWEEN LOW AND HIGH RESOLUTION

In Figure 24 and Figure 25, we visualize the difference between low and high-resolution renderings in terms of noise analysis and smart camera movement. We observe that the contrast of clean score between clean and noisy regions is lower for the lower resolution, but there is still a difference between the noisy and non-noisy regions, resulting in the camera moving in the right direction. Therefore, we always use low-resolution rendering during Smart Camera Movement iterations.

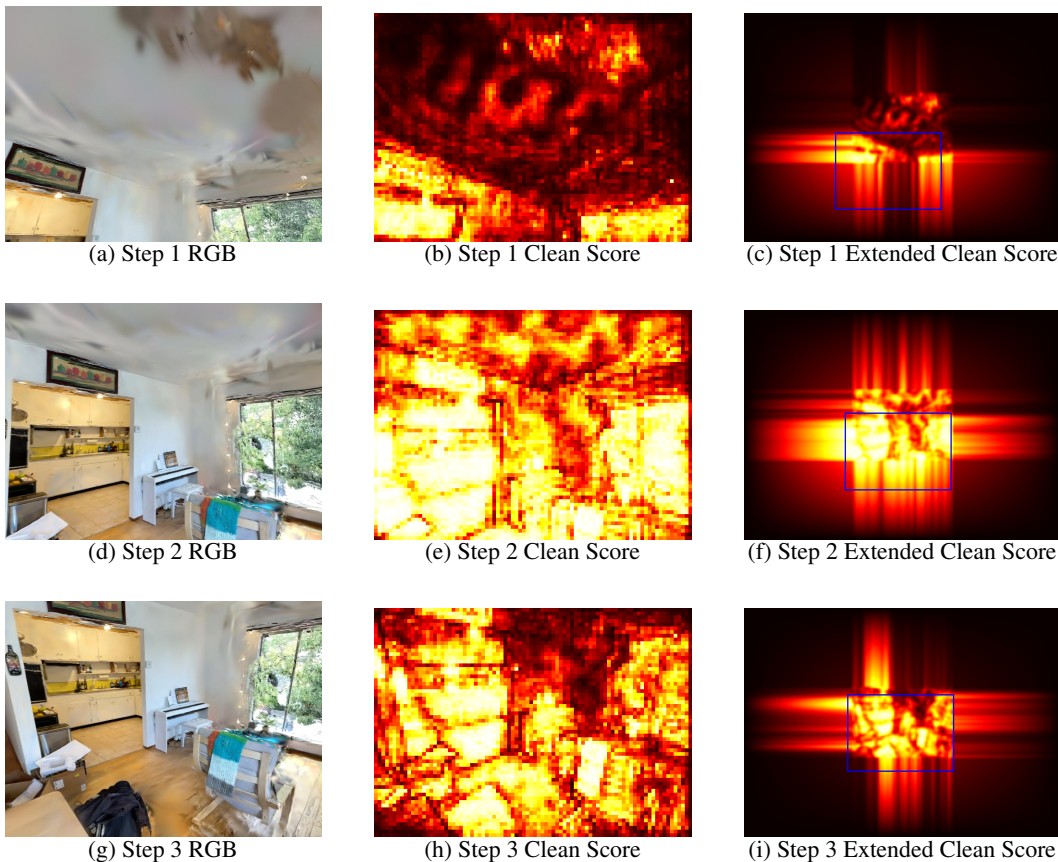

(a) Step 1 RGB     (b) Step 1 Clean Score     (c) Step 1 Extended Clean Score

(d) Step 2 RGB     (e) Step 2 Clean Score     (f) Step 2 Extended Clean Score

(g) Step 3 RGB     (h) Step 3 Clean Score     (i) Step 3 Extended Clean Score

Figure 22: Smart Camera steps for the bouquet scene in different views: RGB, Clean Score, and Extended Clean Score.

### C.3 FAILED RETRIEVAL VISUALIZATIONS

We demonstrated successful retrievals in the main paper. In this section, we demonstrate additional retrieval failure cases. We observe that our framework generally avoids unreasonable matches. Most incorrect matches are due to inherent difficulties and ambiguities in object labeling. Figure 26 visualizes these failure cases. In the first row, the prompt *table* from the bouquet scene matched with a table in the donuts scene. This is a common issue with objects that appear in multiple scenes but are labeled only once. In the second row, the *lamp shade* prompt retrieved the back of a camera flash, which somewhat resembles a lamp shade. In the third row, the "incorrect" retrieval seems more reasonable as the *sheep* in the groundtruth scene is very small. In the fourth row, the *foam darts* prompt was misunderstood by the VLM, retrieving a completely unrelated item. Even the top match from the groundtruth scene was incorrect, as the *foam darts* are small foam bullets for a NeRF gun, as shown in Figure 26(e). We anticipate that our solution's ability to understand foreign objects will improve with future VLM advancements. However, additional research is needed to address ambiguities.

### C.4 LLaVA CAPTION AND RETRIEVAL VISUALIZATIONS

In Figure 27, we visualize the retrieved scene (image) given LLaVA captions as queries. As shown in the figure, the retrieved image matches closely with the image used to generate the caption for the success case. We visualize the failure cases in Figure 28, the failure cases show that the query is misinterpreted by the XDecoder model. Note that the LLaVA caption itself can also be inaccurate, for example, LLaVA referred to the donuts as orange.

### C.5 RANDOM VIEWPOINT RENDERING VISUALIZATION

We visualize the random noise from pre-trained NGRs in Figure 29. Notice that across very different scenes and datasets, the noise pattern is very similar (flat cloud region or elongated Gaussian blobs).

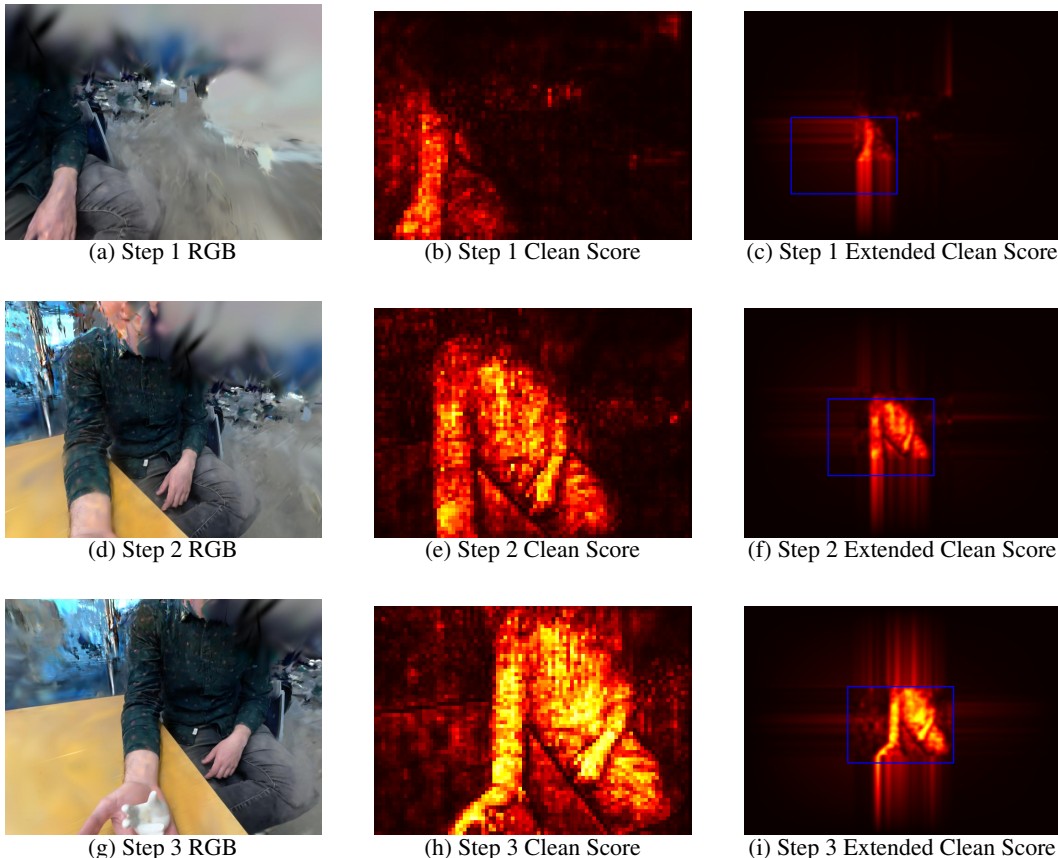

(a) Step 1 RGB          (b) Step 1 Clean Score          (c) Step 1 Extended Clean Score

(d) Step 2 RGB          (e) Step 2 Clean Score          (f) Step 2 Extended Clean Score

(g) Step 3 RGB          (h) Step 3 Clean Score          (i) Step 3 Extended Clean Score

Figure 23: Smart Camera steps for the handhand scene in different views: RGB, Clean Score, and Extended Clean Score.

Existing works evaluate novel view synthesis at viewpoints that are not far from the training image viewpoints. For example, in previous studies, the quality of ScanNet novel view rendering is assessed by using every 8th image in the video sequence for evaluation, with all other images used for training. This approach results in evaluation images that are very close to the training images in terms of viewpoint (Dai et al., 2017; Lao et al., 2023; Bian et al., 2022). Nerfbusters identified this issue and proposed evaluating based on images captured from a second trajectory, different from the training image trajectory. However, even with 3D diffusion modules designed to remove artifacts, the rendering quality remains significantly lower than when rendering viewpoints along the training image trajectory (Warburg et al., 2023). Although the second trajectory in Nerfbusters captures different paths, it still includes similar areas of the scene. Our approach, using completely randomly chosen viewpoints, is markedly different from the training images.

Some NGR works utilize diffusion models for 3D object generation from text or images or to enhance the quality of 3D models given extremely sparse image captures (Liu et al., 2023d; Poole et al., 2022; Zou et al., 2023b). We do not evaluate these methods, as the majority of NGRs do not employ them, and these diffusion models tend to hallucinate the 3D scene rather than accurately capture it (Zou et al., 2023b; Chen et al., 2023). In the datasets used by our work, each scene contains hundreds to thousands of training images. This distinguishes our training setup from those focusing on sparsity. The noise artifacts we observe are due more to incomplete coverage than to training image sparsity, which we believe is an important issue for real-world use and understanding of NGRs not receiving sufficient attention (Warburg et al., 2023).

## C.6 NOISE ANALYSIS AND CAMERA MOVEMENT IN SPLATFACTO AND NERFACTO COMPARISON

In Figure 30, we present additional visualizations of noise analysis and camera movement behavior for Splatfacto and Nerfacto using the same noise distribution. Specifically, we employ the noise Gaussian distribution trained on Splatfacto noise and evaluate it on renderings from the same viewpoints for both Splatfacto and Nerfacto.

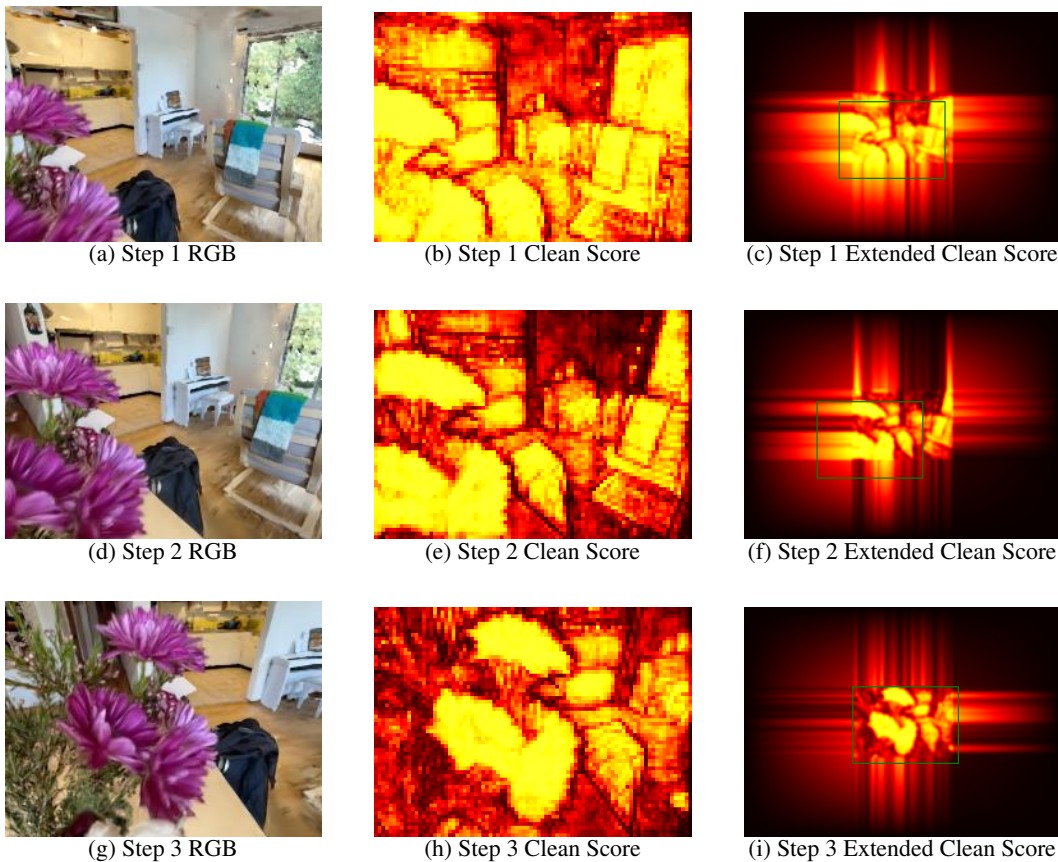

Figure 24: Smart Camera steps low resolution: RGB, Clean Score, and Extended Clean Score.

We observe two key results. First, the noise distribution consistently assigns significantly higher clean scores to clean areas compared to noisy ones, irrespective of whether the noise originates from Splatfacto or Nerfacto. Second, the noise distribution distinguishes slightly more sharply between Splatfacto noise and clean content than between Nerfacto noise and clean content. While this difference is observable, it is minor and has minimal impact on the eventual camera movement.

### C.7    COMPLEX PROMPTS AND SCENE VISUALIZATION

In Figures 31, 32, 33, 34 and 35, we provide additional visualizations for different types of prompts. Specifically, Figure 31 focuses on complex prompts describing subtle relationships between objects (e.g., "a group of," "near," "attached to"). The query in Figure 31(a) also incorporates an environmental style ("A garden"), which is distinct from the rest of the LERF scenes, highlighting the VLM's ability to generalize to diverse contexts. For the query in Figure 31(d), we demonstrate the retrieval result when the object is partially occluded. In this case, the VLM successfully matches the text query with an unoccluded viewpoint of the object as the top choice, while an occluded viewpoint is matched as a non-top choice. This demonstrates the VLM's ability to handle object occlusion to some extent but also highlights the importance of providing high-quality renderings from multiple viewpoints through SCMM.

Figure 32 presents visualizations for retrieving glass objects, which exhibit distinct lighting properties compared to most other materials. Figure 35 retrieves "bright sunlight", which is a unique environmental style and lighting condition that is applicable to outdoor scenes or scenes with windows. Figure 33 illustrates retrieval results for the prompt "A wood object," showcasing the system's capability to identify various objects composed of a specific material. Lastly, Figure 34 evaluates the system's performance with the prompt "A round object," focusing solely on object shape. We note that not all retrieved results are correct—for example, some "glass-like" objects, such as plastic cups, are retrieved. These examples highlight challenges in resolving certain complex associations, which could benefit from further advancements in VLMs. Retri3D, as a framework, is well-positioned to take advantage of such advancements as more powerful VLMs become available.

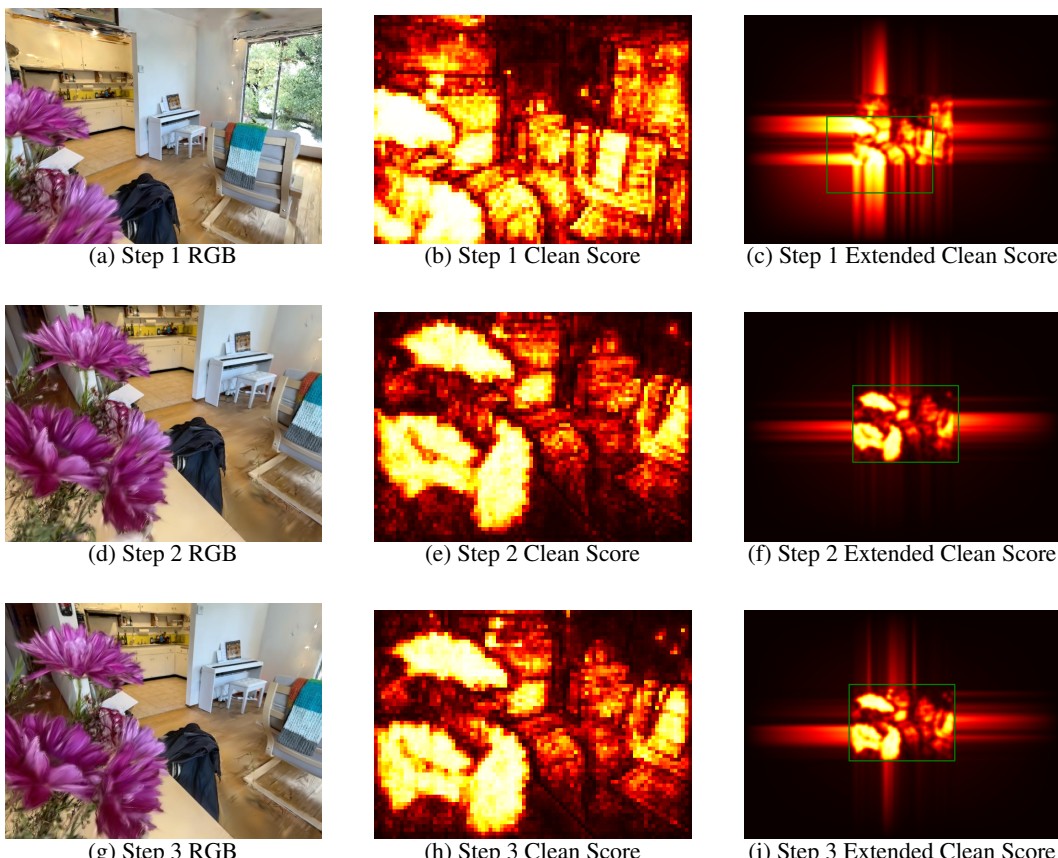

(a) Step 1 RGB     (b) Step 1 Clean Score     (c) Step 1 Extended Clean Score

(d) Step 2 RGB     (e) Step 2 Clean Score     (f) Step 2 Extended Clean Score

(g) Step 3 RGB     (h) Step 3 Clean Score     (i) Step 3 Extended Clean Score

Figure 25: Smart Camera steps for high resolution: RGB, Clean Score, and Extended Clean Score.

In summary, the VLM generally succeeds in associating complex queries with the objects described in the images, effectively utilizing the renderings provided by Retri3D's SCMM module. However, certain cases reveal limitations, such as the misclassification of visually similar objects. The resolution of these limitations could benefit from continued development in VLMs.

## C.8 CLEAN SCENE SCMM AND RANDOM VIEWPOINT BEHAVIOR COMPARISON

In Figure 36, we present the blender lego scene as an example of a well-covered scene without significant floater noise artifacts. While floater artifacts are minimal due to the high-quality training image coverage, we demonstrate that SCMM still offers advantages over random viewpoint sampling. Specifically, random viewpoint sampling may occasionally generate views that are extremely close to the object or even inside it. Such views can cause the rendered object to appear "noise-like" because Gaussian blobs, which are typically trained on external perspectives of objects, are not optimized to represent these unconventional viewpoints.

SCMM mitigates this issue by guiding the camera to viewpoints that are more suitable for rendering and analysis, avoiding positions where Gaussian blobs fail to represent the scene effectively. This results in renderings that maintain higher visual quality and semantic consistency, even in scenes that are otherwise well-covered and free from floater artifacts. This example highlights how SCMM enhances the robustness of rendering and retrieval processes, particularly in challenging scenarios involving suboptimal camera positions.

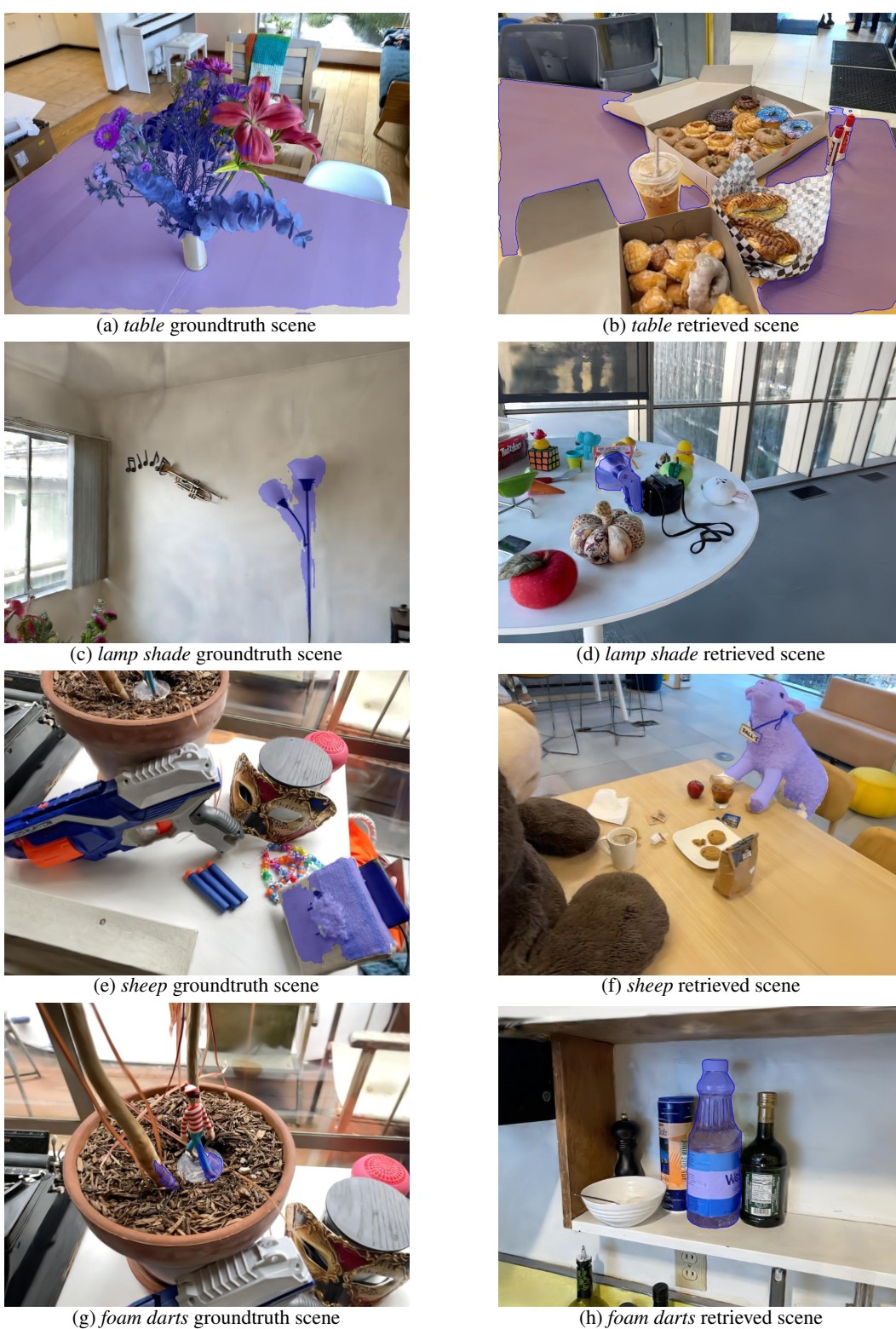

(a) *table* groundtruth scene

(b) *table* retrieved scene

(c) *lamp shade* groundtruth scene

(d) *lamp shade* retrieved scene

(e) *sheep* groundtruth scene

(f) *sheep* retrieved scene

(g) *foam darts* groundtruth scene

(h) *foam darts* retrieved scene

Figure 26: Failed retrievals from LERF dataset. Groundtruth scene containing the label vs. Retrieved scene.

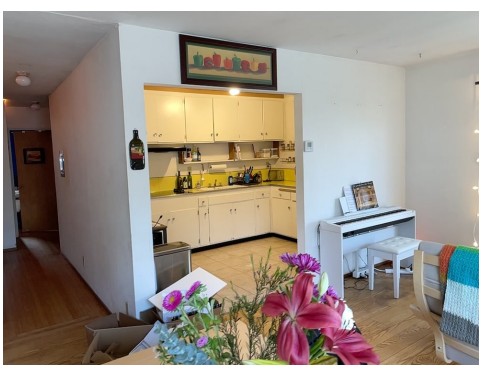 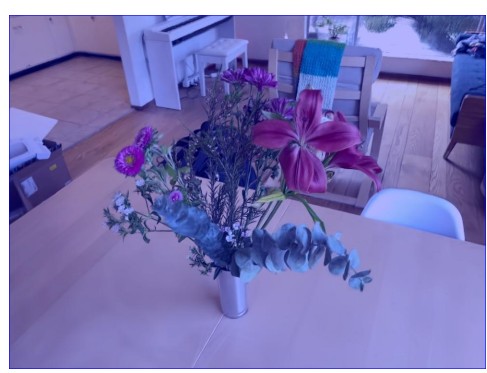

(Success) Query: A kitchen with a white piano, a table with a vase of flowers, and a chair.

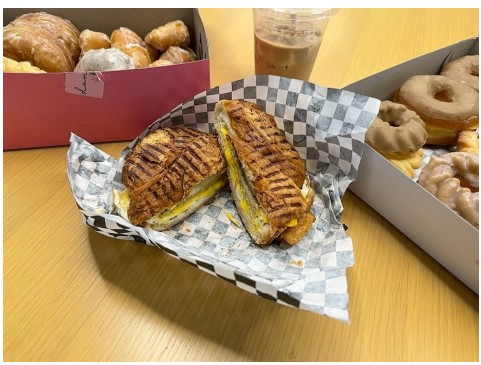 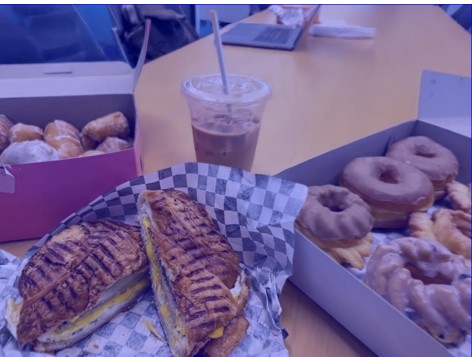

(Success) Query: A table with a sandwich and a cup of coffee on it, along with a box of donuts.

Figure 27: Successful retrievals with LLAVA. Left is the image that generated the LLaVA caption. Right is the top match image from the retrieved scene. The blue mask over the entire image indicates the whole image embedding is matched.

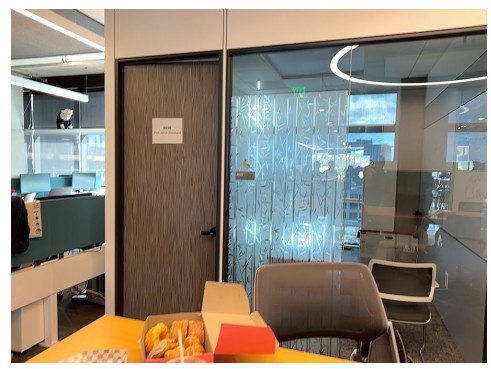 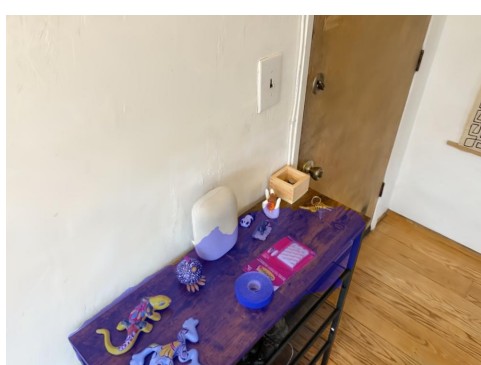

(Fail) Query: A room with a glass door and a desk with a box of oranges on it.

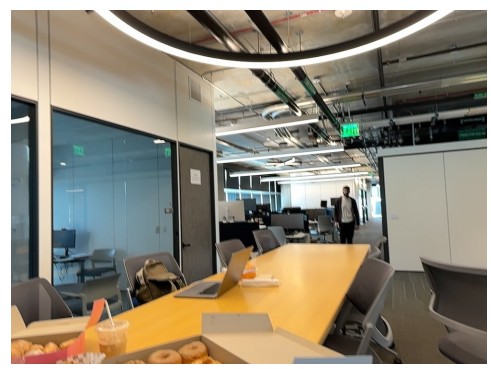 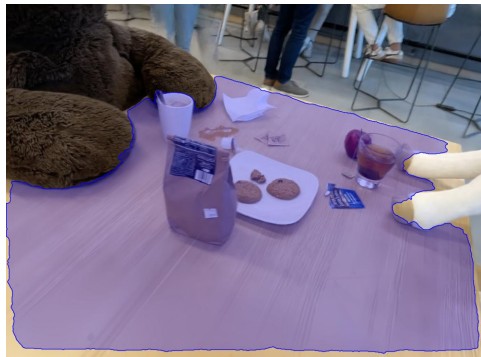

(Fail) Query: A man is walking through a large room with a table full of donuts and a laptop.

Figure 28: Failed retrievals with LLAVA. Left is the image that generated the LLaVA caption. Right is the top match image from the retrieved scene. The blue mask over a section of the image indicates the section image embedding is matched.

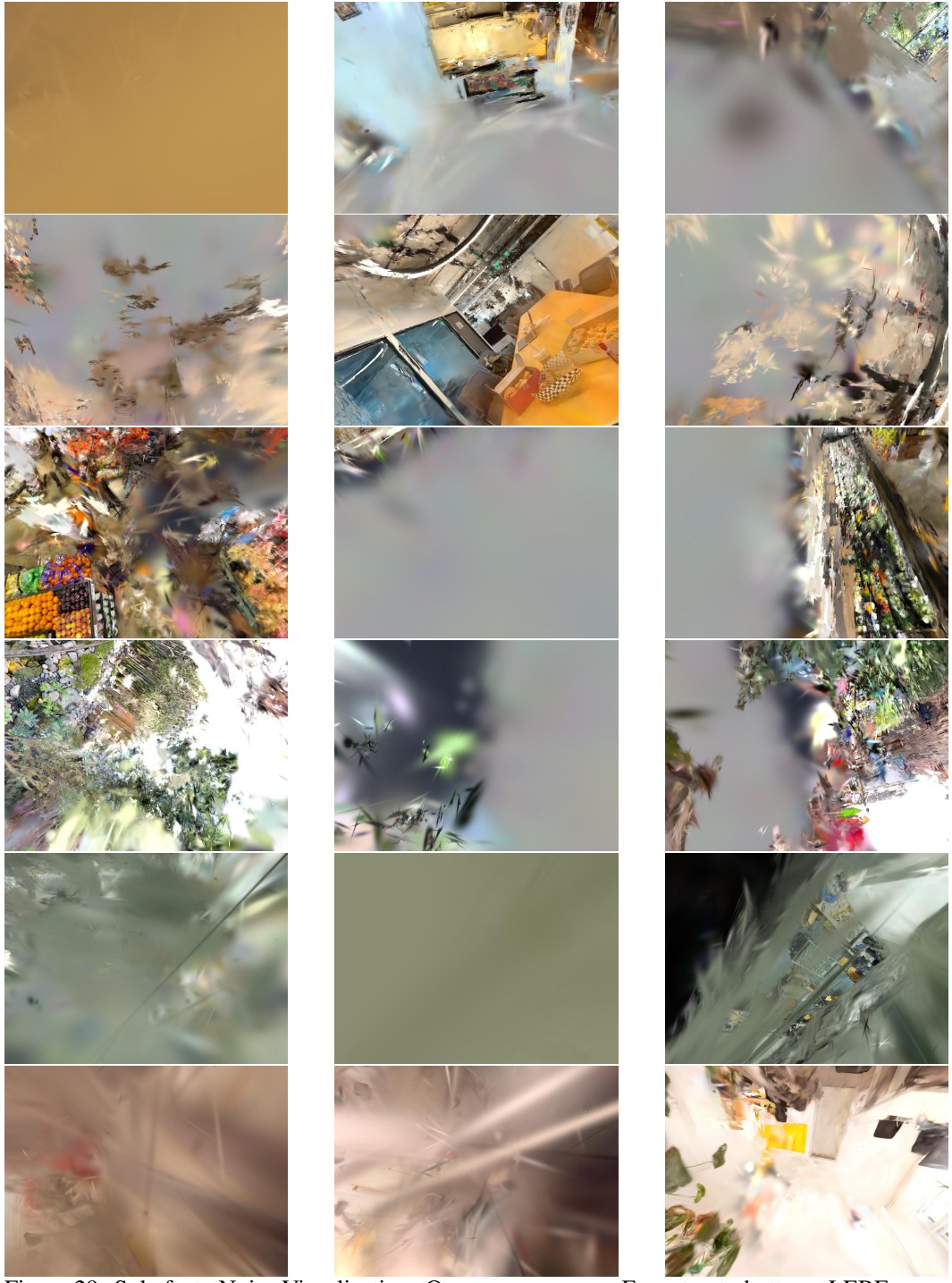

Figure 29: Splatfacto Noise Visualization. One scene per row. From top to bottom: LERF scene *bouquet*, *donuts*, *fruit_aisle*, *sunnyside*; ScanNet++ scene *036bce3393*, *6cc2231b9c*

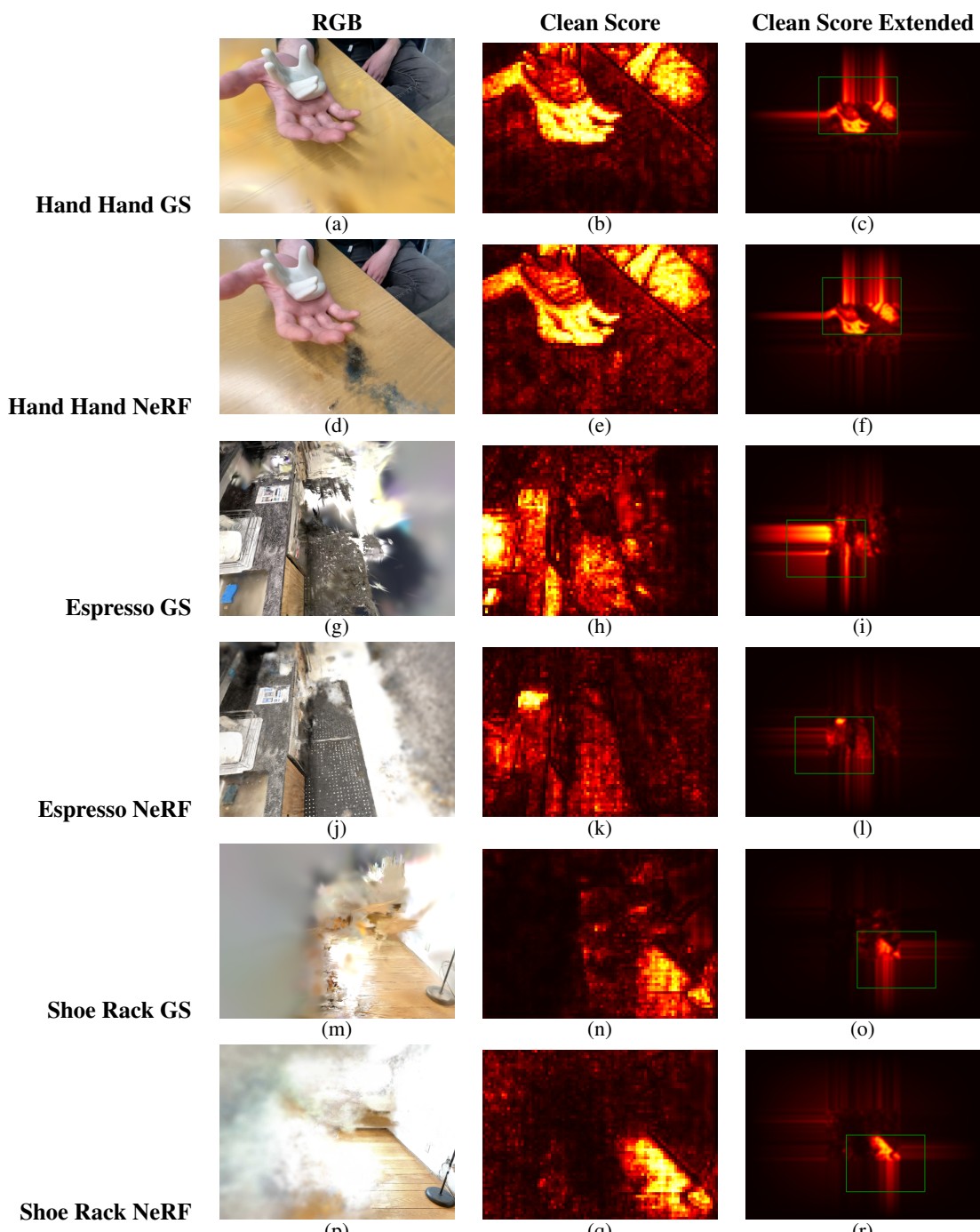

Figure 30: Comparison of Noise Analysis and Camera Movement for Splatfacto and Nerfacto. We note that the location of noise occurrence can be different due to different models and initializations. We observe that the noise between Splatfacto and Nerfacto can be similar, and more importantly, the noise Gaussian Distribution trained with Splatfacto noise can direct the camera to cleaner areas in both Splatfacto and Nerfacto.

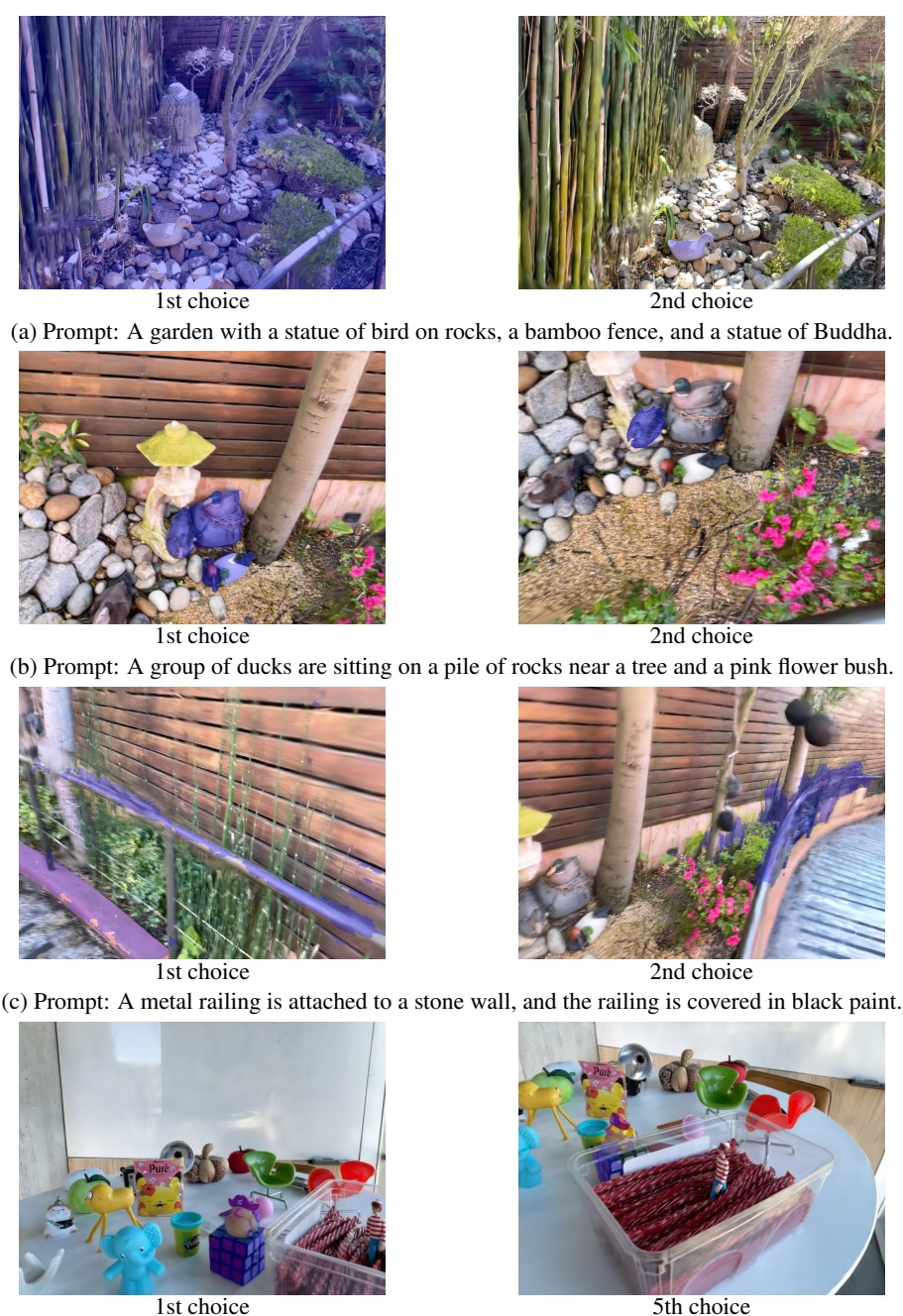

1st choice · 2nd choice

(a) Prompt: A garden with a statue of bird on rocks, a bamboo fence, and a statue of Buddha.

1st choice · 2nd choice

(b) Prompt: A group of ducks are sitting on a pile of rocks near a tree and a pink flower bush.

1st choice · 2nd choice

(c) Prompt: A metal railing is attached to a stone wall, and the railing is covered in black paint.

1st choice · 5th choice

(d) Prompt: A duck on a cube.

Figure 31: Visualization of complex object relations within more intricate scenes. Prompts (a, b, c) are generated by LLaVA, while prompt (d) is hand-designed. (a) Given a complex prompt describing multiple objects, the XDecoder VLM successfully identifies the associated scene. The top choice corresponds to the whole image embedding (hence the mask is associated with the entire image), while the second choice focuses on the bird object in the scene. (b) For a prompt involving complex object relations such as "group of" and "near," the VLM identifies multiple viewpoints associated with the prompt. Since the segmentation masks are created during scene analysis (prior to prompt availability), they may not perfectly align with the prompt's description. (c) Although the LLaVA caption's object association "attached" may not be the most suitable description, the VLM still identifies images closely related to the prompt. (d) The VLM successfully identifies the duck on a cube from a cluster of objects. Its top choices highlight viewpoints where both the duck and the cube are clearly visible. Additionally, we present a lower-ranked choice (d, right), where the VLM identifies the combination of the duck and cube from an occluded perspective. The term "cube" was hand-designed, as we found that XDecoder does not recognize "Rubik's Cube."

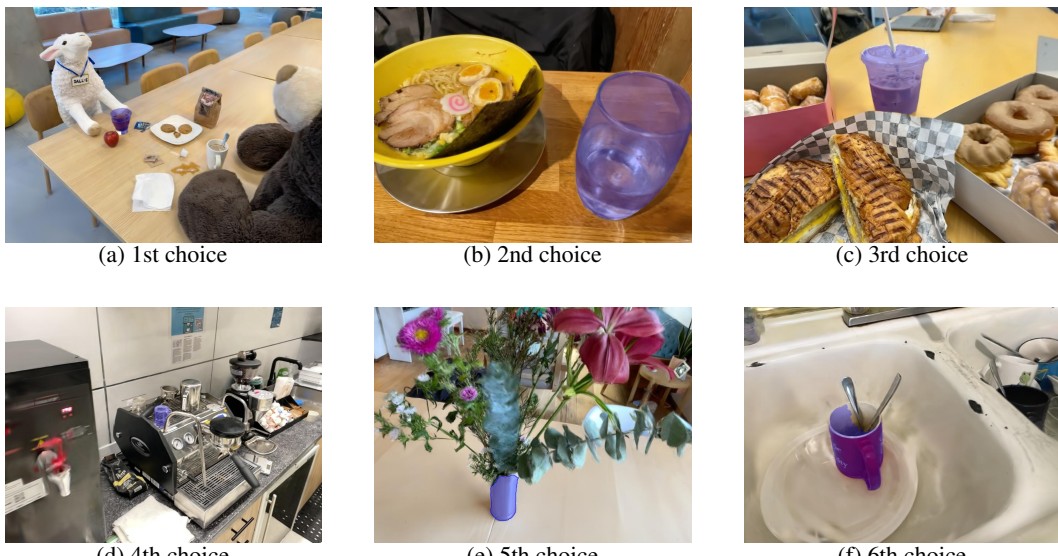

(a) 1st choice     (b) 2nd choice     (c) 3rd choice

(d) 4th choice     (e) 5th choice     (f) 6th choice

Figure 32: Top choices for the prompt "A glass mug," with repeated associations to the same object within the scene ignored. Strictly speaking, the 1st and 4th choices are glass mugs. However, objects similar to glass mugs, such as the glass cup in the 2nd choice and the (likely) plastic cup in the 3rd choice, are selected first as they are more easily identifiable.

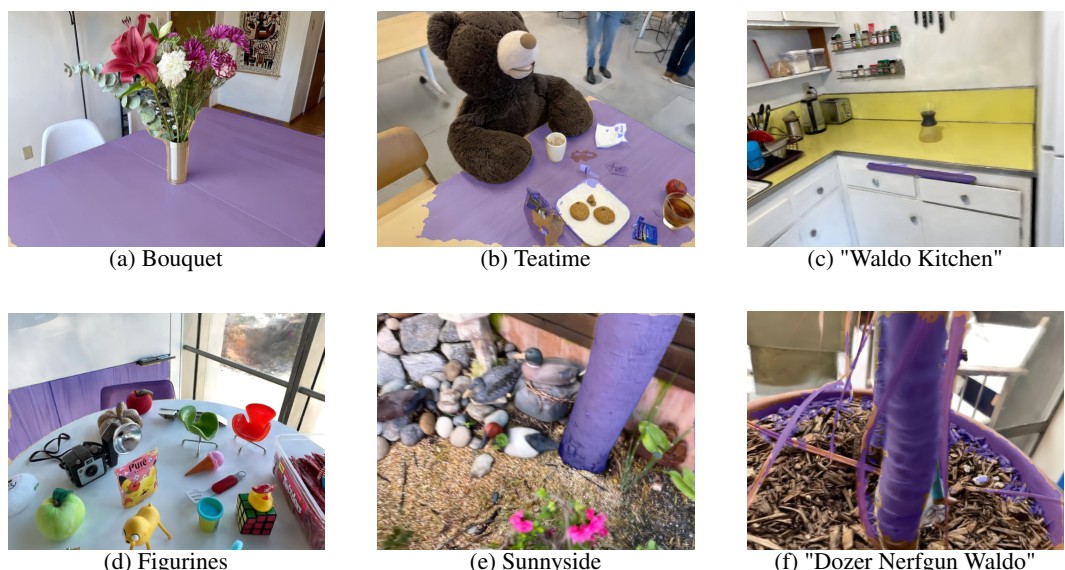

(a) Bouquet     (b) Teatime     (c) "Waldo Kitchen"

(d) Figurines     (e) Sunnyside     (f) "Dozer Nerfgun Waldo"

Figure 33: Selected retrievals for the prompt "A wood object" from different scenes. We observe that the VLM successfully identifies various wood objects as well as wood-like objects, such as the table in "Dozer Nerfgun Waldo."

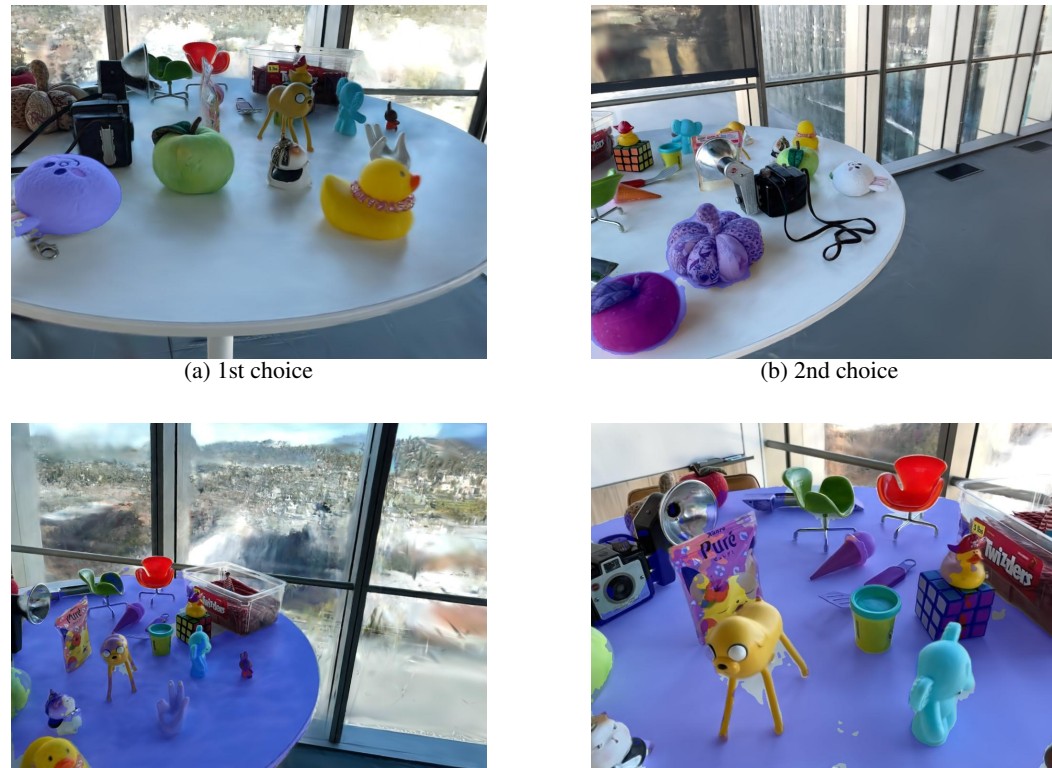

(a) 1st choice
(b) 2nd choice

(c) 3rd choice
(d) 4th choice

Figure 34: Retrievals in the scene "figurine" given the prompt "A round object." The scene "figurine" is used to demonstrate the retrieval of a specific shape (round). While the scene contains multiple round objects, and the VLM successfully identifies most of them, the system also misses some round objects, such as the green apple.

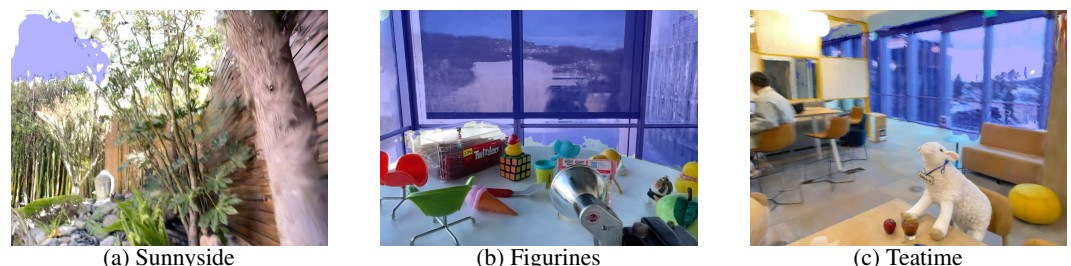

(a) Sunnyside
(b) Figurines
(c) Teatime

Figure 35: The retrieval results for the prompt "Bright sunlight." Retrieved images either contain outdoor sky, or windowed view of outdoor sky.

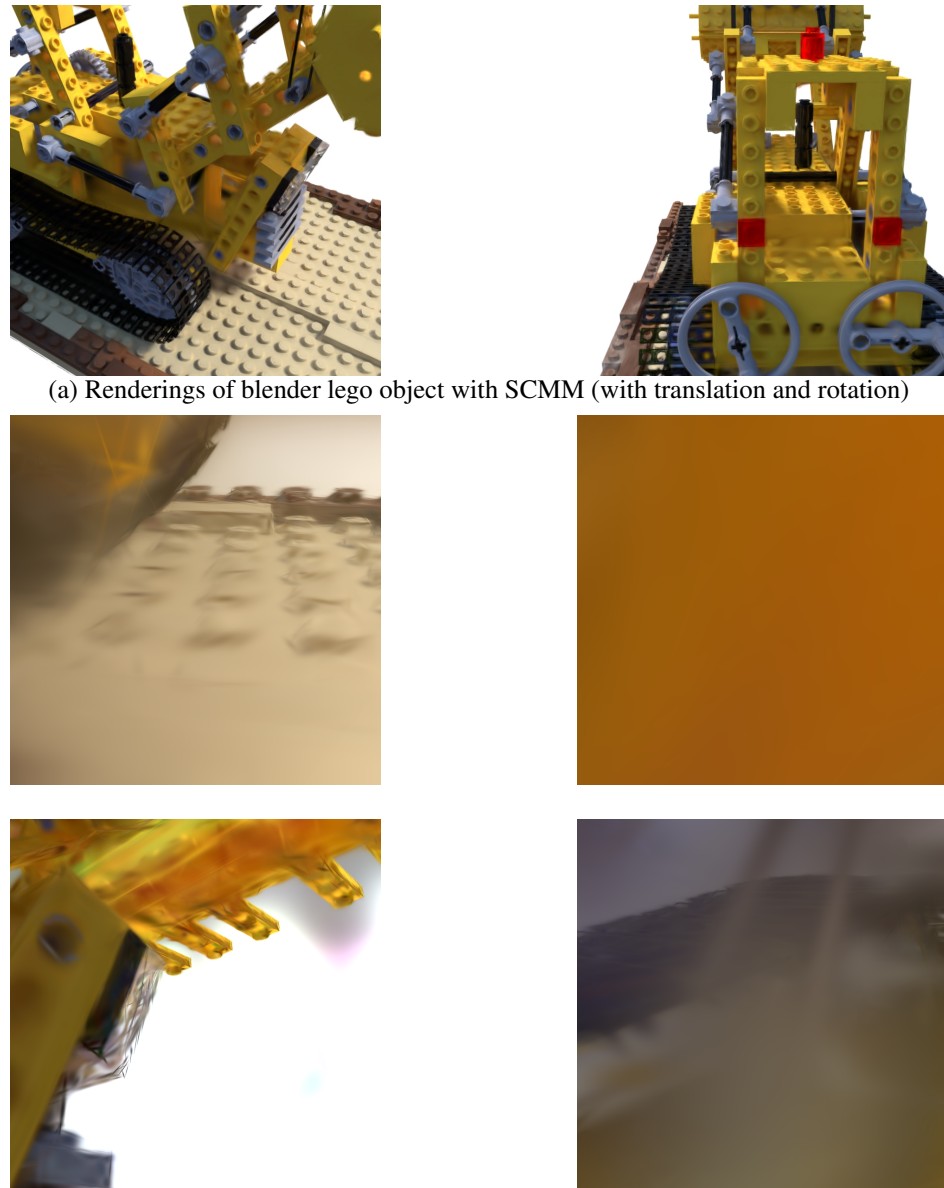

(a) Renderings of blender lego object with SCMM (with translation and rotation)

(b) Renderings of blender lego object with random

Figure 36: Renderings from SCMM and random in an almost noise-free scene (nerf-synthetic blender lego). Due to sufficient training image coverage, floater artifacts are minimal in the lego scene. However, random viewpoint rendering may capture the object from undesirable angles, causing the object itself to appear "noise-like."

