# OpenReview forum: "Retri3D: 3D Neural Graphics Representation Retrieval"
_ICLR.cc/2025/Conference — ICLR 2025 Spotlight_

### Official Review · Reviewer_aXbV · 2024-10-23

**Soundness:** 4
**Presentation:** 4
**Contribution:** 4
**Rating:** 8
**Confidence:** 3

**Summary:**

This paper presents Retri3D, a novel framework for retrieving 3D Neural Graphics Representations (3D NGRs) using text queries. The system leverages cosine similarity between embeddings generated by a pretrained Visual Language Model (VLM) from text queries and RGB renderings of 3D scenes. To enhance the quality of visual feature embeddings, Retri3D introduces two key techniques: Neural Field Artifact Analysis, which uses a multivariate Gaussian model to differentiate clean from noisy pixels using activation maps of VLM, and a Smart Camera Movement Module that iteratively samples new camera angles to reduce noise. Experiments show that Retri3D excels in accuracy, training time, embedding size, and retrieval performance on the LERF and ScanNet++ datasets, outperforming existing baselines.

**Strengths:**

1. This paper addresses a novel problem in retrieving 3D NGRs that previous works have not covered.

2. Retri3D is the first framework capable of efficiently retrieving 3D scenes from large datasets using text queries without requiring training views or camera poses.

3. The proposed Artifact Analysis effectively distinguishes clean and noisy regions in RGB renderings, while the Smart Camera Movement Module identifies cleaner viewpoints.

4. Extensive experiments validate that Retri3D generates accurate embeddings and retrieves 3D scenes efficiently, utilizing moderate storage and training time compared to baseline methods.

In conclusion, the paper's contributions are significant, establishing a novel framework for 3D NGR retrieval. The introduction of two innovative modules enhances embedding quality, and the experiments comprehensively evaluate retrieval accuracy, computational efficiency, and scene coverage, clearly demonstrating advantages over baselines that integrate language features into scene representation.

**Weaknesses:**

1. The experiments focus on only two types of text queries (object labels and LLaVA-generated queries). Testing more complex queries—such as those describing object shapes, textures, environmental styles, materials, lighting, and specific object arrangements—could further reveal the retrieval limits.

2. While the authors evaluate two NGRs (Splatfacto and Nerfacto), many other NGRs may present different artifact patterns, potentially impacting the neural graphics noise analysis. Additionally, noisy or blurred regions could be more pronounced with fewer training views, especially in few-shot reconstruction settings or if the model underfits, which may hinder the noise analysis module's effectiveness in new scenes.

3. The framework's understanding of 3D scenes relies on 2D renderings, treating embeddings from different viewpoints independently. This approach may fail to answer queries about 3D-specific structures or details. Furthermore, since the current retrieval operates at a uniform resolution, querying localized small areas could result in inaccuracies due to the limitations of 2D rendering resolution.

**Questions:**

1. Do tightly clustered noisy features consistently affect all VLMs, and how might performance vary with different architectures?

2. Are the results only applicable to indoor scenes, or does the framework also extend to complex outdoor environments?

3. How are initial cameras set up for a given NGR, and what happens if the initial pose is poorly sampled?

4. As the Smart Camera Movement Module seeks cleaner viewpoints, how does it ensure comprehensive coverage of the entire 3D scene? Will some important parts, such as a corner in the scene, be neglected in all the sampled viewpoints?

5. What field of view (FOV) is used for rendering RGB images, and could wider FOV (such as a fisheye lens or a panorama image) improve feature capture in 3D scenes?

---

> ### Author Response · Authors · 2024-11-21
>
> **W1 [Complex Queries]**
>
> In Appendix C.7, Figures 31–35 of the updated PDF, we have included additional visualizations of the retrieval results for queries focusing on different aspects. We observed that when provided clean views with sufficient coverage by SCMM, the VLM generally performs well in retrieving content relevant to the descriptions.
>
> **W2 [Noise Pattern and Noise Analysis in Few-shot Setting]**
>
> We agree that noise patterns may vary across different NGR architectures. Our findings show that the noise patterns in Splatfacto and Nerfacto are sufficiently similar to enable the use of a shared noise model for effective smart camera movement. Additional visualizations have been added in Figure 30, demonstrating that the noise analysis and camera movement behavior remain very similar when the same noise distribution is applied to handle noise from both Splatfacto and Nerfacto. If other NGRs exhibit distinct noise patterns, a new noise Gaussian model can be fitted and used with Retri3D without requiring further modifications.
>
> For extreme noise in few-shot reconstruction settings, we expect the noise analysis module to remain effective, as renderings with almost complete noise are abundant in the current setting. However, identifying areas in the scene that provide reasonable-quality renderings becomes more challenging. In such cases, an increased number of camera samples may be necessary. For a visualization of noise analysis behavior under conditions of nearly complete noise, please refer to Figure 11(c).
>
> **W3 [3D-specific Information]**
>
> We agree with the reviewer that in cases where 3D-specific information is part of the model but not observable in 2D, such as with 3D CAD model designs, our approach can lead to information loss and should be complemented by other 3D-specific models. In some cases, however, 3D-specific information can also be represented in 2D if the chosen viewpoint captures it effectively. For instance, in Figure 31 (b, c, d), we illustrate several complex object relationships, such as “a group of,” “near,” “attached to,” and “on.” While these terms describe object relationships in the 3D world, the same relationships can still be understood in 2D when the objects are viewed from an appropriate direction.
>
> **W3 [Small Areas and Small Objects]**
>
> The primary VLM we used, XDecoder, is capable of segmenting objects within an image and, theoretically, can identify small objects in renderings. For example, the glass mug in Figure 32(d) is a very small object located in a highly clustered area. If retrieving small areas is the primary concern, the SCMM algorithm could potentially be modified to move the camera and zoom in on individual small objects as they are detected. This approach would provide a more detailed, close-up view of these small objects.

---

> ### Author Response · Authors · 2024-11-21
>
> **Q1 [VLMs and Noise Features]**
>
> We observed that noise features consistently impact modern VLMs trained to generalize across a broad range of image data. This was demonstrated using both the convolution-based vision encoder from XDecoder and the transformer-based vision encoder from OpenCLIP. Models that are less well-trained or have been exposed to a narrower variety of image data during training are likely to perform worse.
>
>
> **Q2 [Complex Outdoor Environments]**
>
> Retri3D does not require special handling for complex outdoor environments. However, if the environment is very large or contains more content, a greater number of renderings would be needed to capture different aspects of the scene. One of the scenes, Sunnyside from LERF, is an outdoor setting that captures a garden with many different objects. Capturing this scene involved a person walking along a path in the clustered garden, making it more complex and information-dense. We present retrieval examples from this scene in Figure 31 (a, b, c).
>
>
> **Q3 [Initial Camera Setup and Poorly Sampled Initial Pose]**
>
> We initialize the smart camera movement module by randomly sampling the location from a Normal distribution and the viewing direction from a uniform distribution. The Normal distribution sufficiently distributes the initial cameras, with a bias for more center views which is more likely to have partially noise-free views. SCMM is capable of rotating to a clean view, even when starting from a viewpoint where only a small section is noise-free (as demonstrated in Figures 23 and 24). Combined with uniform sampling of the viewing direction, this approach is more likely to provide different angles of the objects in the center while also capturing objects in the periphery. This approach provides good coverage of the scenes, while enabling the cameras to navigate to more noise-free views. Additional discussion and details on this are provided in Appendix A.4.1.
>
> If the initial pose is poorly sampled, our SCMM (with rotation) can adjust the viewpoint by rotating to recover a good view, provided that a clean view is accessible from the same camera origin with a new viewing direction. However, if the camera origin is surrounded by noise and no view direction offers a good view, SCMM (with rotation and translation) will be required to recover a suitable viewpoint.
>
>
> **Q4 [Comprehensive Coverage and Corners]**
>
> We enhance the visual semantic diversity of the coverage by selecting the top k dissimilar clean renderings (discussed in Appendix A.4). Specifically, low-resolution renderings chosen for further high-resolution rendering and analysis must have cosine similarities below 0.9. This approach prevents the selection of overly similar renderings. Since the smart camera occasionally converges to similar viewpoints despite starting from different initializations, this process helps improve view variability.
>
> With SCMM, we aim to maximize coverage while still ensuring less noisy rendering. This is because even if we used a very noisy rendering of a corner, it would not be useful for retrieval.
>
>
> **Q5 [FOV]**
>
> We use the FOV consistent with the training images to ensure a consistent FOV is used for comparison of retrieval accuracy among training viewpoint renderings, SCMM, and random sampling. The rendering quality should remain similar if a different FOV is used, thus the retrieval performance should be minimally affected by the specific FOV chosen.
>
> Using FishEye or Panorama projections may introduce unwanted distortions, as existing NGRs typically assume a pinhole camera model. Increasing the FOV while maintaining the pinhole camera model could be a more practical approach, offering the advantage of greater coverage but potentially sacrificing detail. Whether increasing the FOV is overall beneficial likely depends on the specific scene and the focus of the user’s text queries.

---

> ### Comment · Reviewer_aXbV · 2024-11-26
>
> I appreciate the authors' response and the additional experiments that address parts of my concerns. I maintain the acceptance rating.

---

### Official Review · Reviewer_eN88 · 2024-10-27

**Soundness:** 3
**Presentation:** 2
**Contribution:** 2
**Rating:** 6
**Confidence:** 4

**Summary:**

This paper aims to solve a 3D radiance field retrieval problem using text inputs and queries from the dataset constructed by multiview embedding a trained 3D radiance field. The key challenge is to obtain high-quality multiview image embeddings for the database. To solve this problem, the author proposes a noise analysis module to qualify the rendering quality of each rendering view and a camera moving module to guarantee viewpoints that render less noise images. With the powerful Visual-Language model, the method extracts rich information from a constructed database and text embedding for the retrieval task. The authors com

**Strengths:**

1. The overall framework is well-motivated. The authors pay more attention to utilizing the multiview images as the data representation and propose several following designs to conduct the retrieval tasks, which is reasonable.
2. The experiment results are extensive to cover different aspects of the proposed framework as in the main draft and supplementary.

**Weaknesses:**

1. The effectiveness of the smart camera moving module in Tab.1. As shown in Tab.1, using 20 training views almost outperformed every design. In some real applications, storing the radiance field with 20 training views for the following retrieval tasks would still be possible. It only introduces a tiny storage overhead while benefiting the tasks. This makes the setting only suitable if we don't have the training view.
2. The authors propose to quantify the noise in Sec. 3.3 with a heuristic-designed viewpoint selection to determine the noise and clean features, which is counterintuitive for me. It would be better to include more details like how such random viewpoints are generated, and why the random viewpoints will contain valid noise for GMM training.
3. The initial value of the smart camera movement module should also influence the quality. How does it initialize and if we initial from the training views, will the module converge to a bad choice in the test set?

**Questions:**

1. It would be more helpful for the readers to understand why we need a retrieval task by providing detailed applications.
2.  Why not compare with some retrieval method based on conversion 3D representation as listed in lines 45-46? It would also be possible to convert the radiance field representation like 3DGS to a simplified point cloud for this task.

---

> ### Author Response · Authors · 2024-11-21
>
> **W1 [Using Training Views]**
>
> We agree with the reviewer that when training views are available, incorporating a small number of training views may not incur significant storage overhead. However, if these training images do not provide comprehensive coverage of the scene, SCMM can still improve variability in visual semantics by increasing overall scene coverage. For larger or more complex scenes, Retri3D offers the opportunity to render and leverage as many views as needed for accurate retrieval, providing a scalable and efficient mechanism to perform this search. In either case, Retri3D’s retrieval pipeline would still offer advantages in terms of speed and memory efficiency compared to other baseline methods.
>
> However, we observe that training images or related information are not always available. Pre-trained NGRs are increasingly treated as standalone data formats for sharing, as seen on platforms like PolyCam [1] and Luma AI [2]. These platforms typically distribute NGR models without accompanying training images, perhaps to simplify sharing or protect intellectual property or user rights. In such scenarios, our method is particularly valuable, as it enables robust and effective retrieval without requiring training views. This flexibility ensures broad applicability across diverse real-world use cases where training data may not be accessible.
>
> [1] PolyCam: https://poly.cam/
> [2] Luma AI: https://lumalabs.ai/dashboard/captures
>
>
> **W2 [Viewpoint Selection for Noise Gaussian Training]**
>
> We determine if a feature is noise or clean based on its distance to a trained noise Multivariate Gaussian distribution. To train this noise distribution, we generate renderings at randomly sampled viewpoints across multiple scenes. Specifically, the camera origins are sampled uniformly within the 1 × 1 × 1 box of each scene, and the view direction components x, y and z are sampled uniformly between [−1, 1], and the view direction vector is normalized.
>
> The features generated from these renderings are suitable for noise distribution training for the following reasons:
> a. **Noise Dominance in Random Viewpoints**: Renderings produced by this random viewpoint sampling process are dominated by noise content, as illustrated in Figure 29. Thus, the features generated by the VLM from these renderings are primarily noise features.
> b. **Consistency of Noise Patterns**: Second, the noise patterns are repetitive and consistent across different scenes, leading to a noise feature cluster as seen in Figure 2 (Noisy).
> c. **Minimal Influence from Clean Features**: Only few clean features exist and they are not consistent across different scenes.
>
> We use a Multivariate Gaussian distribution that has a single center for noise representation. Since noise features form a single cluster, this distribution is suitable to represent the noise features and provide a strong form of regularization. Note that we do not use GMM (Mixture of Gaussians), as it will introduce multiple centers, not representative of the single cluster the noise features form. Additionally, the few clean features do not consistently shift the Gaussian center in any specific way, and should have minimal impact on the noise distribution training. Therefore, the Multivariate Gaussian distribution effectively represents the noise.
>
> We have updated Appendix A.2 to incorporate details on the camera sampling for the noise Multivariate Gaussian distribution training.
>
>
> **W3 [Camera Initialization and Initialization from Training Viewpoints]**
>
> We initialize the smart camera movement module by randomly sampling the location from a Normal distribution and the viewing direction from a uniform distribution. The Normal distribution sufficiently distributes the initial cameras, with a bias for more center views which is more likely to have partially noise-free views. SCMM is capable of rotating to a clean view, even when starting from a viewpoint where only a small section is noise-free (as demonstrated in Figures 23 and 24). Combined with uniform sampling of the viewing direction, this approach is more likely to provide different angles of the objects in the center while also capturing objects in the periphery. This approach provides good coverage of the scenes, while enabling the cameras to navigate to more noise-free views. Additional discussion and details on this are provided in Appendix A.4.1.
>
> If the cameras are initialized from the training views, as the training view is mostly noise free, SCMM will not move the camera.

---

> ### Author Response · Authors · 2024-11-21
>
> **Q1 [Retrieval Application]**
>
> The retrieval solution can benefit existing platforms hosting NGRs such as PolyCam and Luma AI, improving user experience for content search. Searching for relevant NGR content also have important downstream applications. Leading softwares in game and art design has started incorporating NGRs [3, 4, 5], necessitating data stores for 3D assets in NGR formats. Additionally, since online NGR stores do not ship with training images, we believe our efficient retrieval solution can be highly beneficial for designers who wish to incorporate pre-existing NGRs into their work.
>
> [3] Luma AI Unreal Engine Plugin: https://www.fab.com/listings/b52460e0-3ace-465e-a378-495a5531e318
> [4] Unity Gaussian Splatting: [https://radiancefields.com/gaussian-splatting-unity-plugin-updated](https://radiancefields.com/gaussian-splatting-unity-plugin-updated)
> [5] Blender Gaussian Splatting: [https://radiancefields.com/editable-gaussian-splatting-in-blender](https://radiancefields.com/editable-gaussian-splatting-in-blender)
>
>
>
> **Q2 [Comparison with Traditional 3D Representation Retrieval]**
>
> We could not compare with the retrieval accuracy by converting to the traditional 3D representations: First, traditional representation 3D retrieval focuses on single 3D objects as opposed to complex scenes. To our knowledge, existing retrieval works for 3D objects cannot be directly applied to point cloud or other traditional representations converted from NGR scenes. Second, as we demonstrate in this work, addressing noise in NGRs is a critical challenge in retrieval that we address with SCMM. This challenge would be exacerbated if the noise in the NGRs is converted into artifacts in point clouds or other representations, where it may not even be identifiable as noise. We attempted to convert NGRs to point clouds and found the quality to be too poor for effective analysis.

---

> ### Comment · Reviewer_eN88 · 2024-11-26
>
> The response solved most of my concerns. According to the author's response, I agree that the proposed method could benefit direct 3D asset retrieval. I will raise my score to 6.

---

### Official Review · Reviewer_qmDY · 2024-11-04

**Soundness:** 4
**Presentation:** 3
**Contribution:** 3
**Rating:** 8
**Confidence:** 5

**Summary:**

The authors propose a novel framework for text-based retrieval of NGR representations (NeRFs, Splats and derivatives).
They approach the problem by utilizing an off-the-shelf VLMs to extract feature embeddings from clean scene renders and match them with query text embeddings. Clean scene renders are obtained by iteratively applying Noise Analysis (using the same VLMs) to identify the direction towards clean scene and using smart camera movements module to converge to a cleaner render.

**Strengths:**

* To my knowledge, the concept of iteratively refining camera positions to achieve cleaner renders appears to be novel.
* The use of SMCC is well-supported by the evidence:
  * The retrieval quality is demonstrated to be significantly higher than previous baselines or random views.
  * Section 4.5 effectively highlights that the smart camera approach offers high coverage of the scene and the training portion of the scene
  * Strong speed / memory benchmarks
* It is well-proven that proposed solution is compatible with various VLMs
* Overall, the paper is concise, and the conclusions are mainly well-supported.

**Weaknesses:**

1. Two major assumptions of the paper are not well-addressed (please refer to the Questions section), namely:
* *noise features remain consistent across different scenes and models*
* *Some content can be noise-free, but they constitute only small portions of the images.*

**Questions:**

1. (line 303) Major claim of the paper:
 > noise features remain consistent across different scenes and **models**

  While it is clear and safe to assume noise features are of the same distribution within the same model family, it's not clear why the same holds for various models, e.g. are NeRFs and Splats features distributed the same? Maybe a mixture of gaussians suit better here? Please discuss.

2. (Claim on line 288) Major claim of the paper:
 > [*...about sampling random view-points in pre-trained NGR scenes...*] Some content can be noise-free, but they constitute only small portions of the images.

  I would expect a robust retrieval pipeline to work on both noisy and clean scenes. For example, how would it handle "ground-truth" scenes? With recent advancements in Splats, I would assume that the percentage of clean renders from random camera viewpoints could be significantly higher. As you train on increasingly clean datasets, SMCC’s performance may degrade. Please discuss how this issue could be mitigated.

3. Could you clarify how cameras are initialized in the SMCC module? If they are initialized randomly, does SMCC consistently converge, or is there a risk of getting stuck in local minima?

---

> ### Author Response · Authors · 2024-11-21
>
> **1. [Noise Across Models]**
>
>  We thank the reviewer for pointing this out and we clarify that noise features are more consistent for the same model, but can be less similar across different models. However, the noise features are still closer to noise features from other models than to the clean features. We have clarified the statement in the paper (page 6, line 305). To evaluate the impact of these differences on the noise analysis and camera movement, we provide additional visualizations in Figure 30. These visualizations compare noise analysis and camera movement when the same noise Gaussian distribution (trained from GS) is applied to handle noise from both GS and NeRF. A summary is provided here and also in Appendix C.6:
>
> In summary, we observe the following:
> i) The noise distribution assigns significantly higher clean scores to actual clean areas compared to noisy areas, whether the noise originates from GS or NeRF.
> ii) The noise distribution creates slightly sharper distinctions between GS noise and clean content than between NeRF noise and clean content. However, this difference is small and has minimal effect on the eventual camera movement.
>
> Applying a mixture of Gaussians could enhance robustness in noise analysis. However, as demonstrated in Table 9, the source noise for the Multivariate Gaussian training has minimal effect on overall retrieval accuracy. Consequently, incorporating a mixture of Gaussians is unlikely to significantly impact retrieval accuracy.
>
> **2. [SCMM on “ground-truth” Scenes]**
>
>  If a scene can be rendered from any viewpoint without introducing noise, SCMM's noise navigation module is not strictly necessary to achieve clean renderings. In such cases, the performance of random sampling can approach that of SCMM. However, it is important to emphasize that SCMM's performance does not degrade under these conditions, as the SCMM still has good coverage of various content of the scene by selecting views that are not very similar, as described by the process in Appendix A.4. Thus, SCMM is a robust solution for both noisy and cleaner scenes. Additionally, Retri3D’s retrieval pipeline with embedding generation and matching is designed to be efficient and effective for scenes of varying noise levels.
>
> In practice, we note that even for simple synthetic scenes that are well-covered by training viewpoints, noisy renderings can still occur. To illustrate this, we use the nerf-synthetic LEGO object as an example of a “clean” scene in Figure 36. This LEGO object is well-covered from all training viewpoints, resulting in minimal floater artifacts. Despite this, renderings can appear “noise-like” if the camera is positioned very close to or inside the object. This happens since neural graphics representations with limited model parameters may not perfectly represent the scene or because some viewpoints may have insufficient coverage in the training data.
>
> **3. [Camera initialization and Convergence Behavior]**
>
>  We initialize the smart camera movement module by randomly sampling the location from a Normal distribution and the viewing direction from a uniform distribution. The Normal distribution sufficiently distributes the initial cameras, with a bias for more center views which is more likely to have partially noise-free views. SCMM is capable of rotating to a clean view, even when starting from a viewpoint where only a small section is noise-free (as demonstrated in Figures 23 and 24). Combined with uniform sampling of the viewing direction, this approach is more likely to provide different angles of the objects in the center while also capturing objects in the periphery. This approach provides good coverage of the scenes, while enabling the cameras to navigate to more noise-free views. Additional discussion and details on this are provided in Appendix A.4.1.
>
> To obtain good renderings from SCMM for retrieval, ideally we want good coverage of the scene, while capturing noise-free views. With SCMM, it is possible for some initial camera placements to fail in navigating to a noise-free view. Thus, an initialization strategy that distributes the initial camera points, increases the chances of identifying cleaner views while providing sufficient coverage.

---

> > ### Comment · Reviewer_qmDY · 2024-11-26
> >
> > I would like to thank the authors for providing detailed clarifications about the major claims of the paper.
> > I am satisfied with these explanations.
> > The proposed method establishes a robust baseline for the problem, offering value to the research community. Therefore, I maintain my "accept" rating.

---

### Official Review · Reviewer_vhd3 · 2024-11-07

**Soundness:** 4
**Presentation:** 3
**Contribution:** 3
**Rating:** 8
**Confidence:** 4

**Summary:**

The paper introduces Retri3D, a novel framework for text-to-3D scene retrieval from repositories of neural graphics representations (NGRs). Retri3D leverages pretrained Vision-Language Models (VLMs), like CLIP, to generate embeddings of both text and rendered images, enabling efficient retrieval across a wide range of 3D scene representations, such as NeRF and 3D Gaussian Splats (3DGS). The core contributions include a Neural Graphics Noise Analysis (NGNA) and a Smart Camera Movement Module (SCMM), which collectively enhance the quality of view selection by detecting and avoiding artifacts, thus improving retrieval accuracy. The system demonstrates compatibility with multiple datasets (LERF, ScanNet++) and achieves superior retrieval speed and storage efficiency.

**Strengths:**

1. Retri3D introduces a novel approach to 3D retrieval by using a two-pronged methodology: noise analysis via Neural Graphics Noise Analysis and selective viewpoint rendering through the Smart Camera Movement Module. These elements enable the retrieval of high-quality embeddings from rendered images given the typical noise in the NGRs.

2. By leveraging pretrained VLMs, Retri3D achieves efficient and accurate retrieval without the need for extensive retraining or dataset-specific tuning, which is an improvement over previous methods. The noise analysis method proposed is demonstrated to outperform traditional NeRF uncertainty estimation techniques, emphasizing the robustness of the framework in achieving high-quality feature extraction under diverse conditions​.

3. The paper is overall well-written and well-organized. The proposed pipeline is clean yet effective, with potential for a significant number of downstream applications.

**Weaknesses:**

I do not have a major concern over the paper. Addressing the following would further improve the quality of this paper:

1. Comparative Analysis Limitations: While Retri3D shows strong results on LERF and ScanNet++ datasets, the paper lacks comparative results with more recent methods such as TIGER[1] or N2F2[2]. Including these would provide a broader context for Retri3D’s advancements and its relative strengths and weaknesses across a more diverse set of models and techniques​. The author should also include ConDense[3] in the related works and have a dedicated discussion/comparison, since it could also be applied to this specific task.

2. Dependency on VLM Embeddings: Retri3D relies heavily on VLMs for generating text and visual embeddings, which inherently limits its performance to the capabilities of the underlying VLM model. This dependency means that Retri3D’s retrieval quality might suffer in cases where the VLM struggles with certain text-visual correlations, particularly in scenes with complex or subtle object relations.

3. Potential Generalizability Issues with Scene Complexity: Although Retri3D demonstrates high performance on scenes with distinct and identifiable objects, its retrieval accuracy in complex scenes (compositional) with overlapping or occluded objects is less explored. A detailed discussion of how Retri3D would handle such scenarios or results comparing its retrieval accuracy in simple versus complex scenes would clarify its effectiveness across varying levels of scene complexity​.

References: [1] Xu, Teng, et al. “TIGER: Text-Instructed 3D Gaussian Retrieval and Coherent Editing.” arXiv preprint arXiv:2405.14455 (2024). [2] Bhalgat, Yash, et al. “N2F2: Hierarchical Scene Understanding with Nested Neural Feature Fields.” arXiv preprint arXiv:2403.10997 (2024). [3] Zhang, Xiaoshuai, et al. "ConDense: Consistent 2D/3D Pre-training for Dense and Sparse Features from Multi-View Images." ECCV 2024.

**Questions:**

1. Adding recent works as mentioned in W1 in the relevant section (related works, experiments, and/or discussions), especially [2] and [3].
2. Adding a dedicated discussion/analysis on the model robustness as mentioned in W2 and W3.
3. The paper presents a robust noise analysis technique, yet an interesting avenue might be replacing NGNA with NeRF uncertainty estimation methods. How would this alternative affect the accuracy, speed, and storage efficiency of Retri3D?

---

> ### Author Response · Authors · 2024-11-21
>
> **1. [Additional Comparisons and References]**
>
> We thank the reviewer for the references to the recent works. Since none of the mentioned works have been open-sourced, we are unable to compare against them directly quantitatively. We have included a qualitative comparison against these works in Section 2.4. We also discuss here in more detail.
>
> TIGER uses a gaussian-based visual embedding field similar to the one proposed in LangSplat. It further enables editability of the Gaussian field using SDS loss of 2D image editing diffusion model and multi-view diffusion model. N2F2 uses a NeRF-based visual embedding field similar to the one proposed in LERF. Its main contribution is using different subsets of the vector sampled from the visual embedding field to represent the visual semantics at different scales. This differs from LERF in using a scale parameter as input to the LERF MLP to implicitly encode visual semantics at varying scales. Overall, all of TIGER, N2F2, LERF and LangSplat requires training a visual embedding field, and render from it to produce dense visual embeddings for retrieval. ConDense does not train a visual embedding field directly, it instead generates a visual embedding field from a 3D transformer that takes in the NeRF feature grid as input. This visual embedding field is further processed through volumetric rendering and compared with 2D visual features for regularization. The dense visual features can be made sparse by training the 2D and 3D backbones to decide the key sparse features, regularized by SuperPoint [1] during training. However, since they have only tested with viewpoints on the original camera trajectory (ConDense Appendix A.3), whether such key feature detection process is still valid from random viewpoints would require further experiments.
>
> When used for efficient retrieval of scenes, TIGER and N2F2 have the same shortcomings as LERF and LangSplat (qualitatively compared against in the paper): i) expensive training of the visual embedding field alongside the RGB NGR, and retraining is needed if a new VLM is used ; ii) expensive volumetric rendering or rasterization through the visual embedding field before the visual embeddings can be compared with the text embeddings for retrieval; iii) the need for the original training images to create the visual semantic field; and iv) the need for storing dense visual features.
>
> ConDense does not incur the computation cost associated with training the visual embedding field per scene. However, it requires 35000 V100 GPU hours and 4000 A100 GPU hours to prepare the NeRF scenes and train the 3D transformer. Moreover, a 3D transformer trained for NeRF may not work for different types of NGRs (e.g, Gaussian Splats) and this is not demonstrated in the paper. ConDense also requires the original camera trajectories for sampling.

---

> ### Author Response · Authors · 2024-11-21
>
> **2. [Dependency on VLM Embeddings and Scene Complexity]**
>
> We have added a discussion and evaluation of this in Appendix C.7. While Retri3D enables identifying clean renderings with high coverage for better matches, Retri3D is reliant on the efficacy of the embeddings generated by the VLM. We have added more visualizations in Figure 31 of the updated version to showcase the system’s performance given more complex queries in more clustered complex scenes. We observe that the system successfully retrieves scenes when queries involve subtle object relations such as “group of,” “near,” and “attached to.” Additionally, we demonstrate an example where “A duck on a cube” is retrieved even when the object pair is partially occluded.
>
> Beyond the VLM’s capability, several design choices enable Retri3D to handle more complex object relationships in complex scenes:
>
> *a. SCMM for High-Quality Viewpoints*
> SCMM allows Retri3D to discover high-quality viewpoints from different angles. In complex scenes, Retri3D can sample diverse, high-quality viewpoints to increase the likelihood of matching complex text queries that may only be evident from specific viewing directions. When VLMs struggle with text-visual associations due to occlusion, Retri3D’s ability to render non-occluded views can lead to more relevant visual embeddings, improving their alignment with text descriptions (as demonstrated in Figure 31(d)).
>
> *b. Multiple Queries for Multi-Object Scenes*
> A complex scene can contain a variety of objects. If the user is aware of them, or the user wants to find a scene that matches several descriptions, the user can use multiple queries simultaneously for retrieval. The scene that is the best match (on average) for most of the queries can be identified, and processing multiple queries can be performed very efficiently with Retri3D (detailed in Appendix B.5, Multiple Queries Retrieval).
>
> *c. Adaptability with More Powerful VLMs*
> Retri3D can flexibly integrate more advanced VLMs to improve system performance in the future. This would only require simple retraining of the noise gaussian, rather than the visual semantic field or the 3D transformer as in the existing works.
>
> **3. [Incorporate other Uncertainty Estimation Methods]**
>
> We compare our noise analysis method with the state-of-the-art NeRF uncertainty estimation method (Bayes’ Rays) in Appendix A.5. We demonstrate that our solution achieves similar performance (better in RGB uncertainty estimation, worse in terms of depth uncertainty estimation) compared to Bayes’ Rays. During our testing, Bayes’ Rays is reasonably fast (about a minute to train on top of pre-trained NeRF). During scene noise analysis, our solution uses VLM’s activation maps, where Bayes’ Rays render additional uncertainty values. We expect using Bayes’ Rays for noise analysis to achieve slightly worse accuracy given that we move cameras based on RGB uncertainty. Overhead for computing and memory is minimal for either solution.
>
> Other solutions can incur very high overhead [2] or require a modification to the underlying NGR model [3], which is not very suitable for our setting.
>
> A major advantage of our solution over Bayes’ Rays is generalizability. Since Bayes’ Rays is a dedicated work that tackles NeRF uncertainty estimation (via NeRF MLP Hessians), it is unclear how it can be applied to Gaussian Splatting. In addition, using the VLM’s activation also introduces fewer components over the existing pre-trained NGR since the VLM is required for feature extraction already.
>
> [1] SuperPoint: Self-Supervised Interest Point Detection and Description, DeTone et al., CVPR 2018
> [2] Density-aware NeRF Ensembles: Quantifying Predictive Uncertainty in Neural Radiance Fields, Sunderhauf et al., ICRA 2022
> [3] Stochastic Neural Radiance Fields: Quantifying Uncertainty in Implicit 3D Representations, Shen et al., 3DV 2021

---

> ### Comment · Reviewer_vhd3 · 2024-12-03
>
> I appreciate the authors' response and the additional discussions added into the manuscript. I will maintain my original score and recommend acceptance of the paper.

---

### Author Response · Authors · 2024-11-21
**Response to All Reviewers**

We thank the reviewers for their valuable feedback. We appreciate that the reviewers found the overall framework useful and recognized the design of the noise analysis and smart camera movement as interesting. Below, we address all comments and we have added new results and explanations to the paper based on the reviewers' feedback.

Specifically, we have updated the following:
- **Section 2.4 Related Work on “3D Neural Graphics Representations Semantic Understanding”**: We provide additional discussions on several related works mentioned by reviewer vhd3.
- **Section 3.3 "Neural Graphics Noise Analysis"**: Further clarifications are included on the noise similarity across different models.
- **Appendix A.4.1 “Camera Initialization and Convergence”**: We expand on the discussion of camera initialization, as raised by the reviewers. Reviewers have also asked about camera convergence, with slightly different focuses. We provide an explanation of the overall convergence behavior in Appendix A.4.1, but we also provide detailed responses to address each reviewer’s question directly in the comments below.
 - **Appendix C.6 “Noise Analysis and Camera Movement in Splatfacto and Nerfacto Comparison” and Figure 30**: We provide visualizations discussing the behavior of noise analysis and SCMM when the same noise distribution is applied to handle Splatfacto and Nerfacto noise.
- **Appendix C.7 “Complex Prompts and Scene Visualization”**: We include visualizations of retrieval results based on the prompt types mentioned by the reviewers in scenes of greater complexity. Please refer to the explanations in Appendix C.7 and the captions for Figures 31 - 35 for further details.
- **Appendix C.8 “Clean Scene SCMM and Random Viewpoint Behavior Comparison”**: We provide a visual comparison of rendering quality with and without SCMM on a clean LEGO scene, demonstrating how SCMM remains useful even when the scene is largely artifact-free. Please refer to the explanations in Appendix C.8 and the caption for Figure 36 for further details.

---

### Meta-Review · Area_Chair_hSvF · 2024-12-21

**Metareview:**

Summary: This paper presents a new framework for text-to-3D scene retrieval from a database of neural graphics representations (e.g., NeRF or 3DGS). It uses pretrained vision-language models to help select clean views rendered from the neural graphics representations to avoid artifacts. This helps improve the retrieval accuracy and storage efficiency.

Strength:
- An interesting task for 3D scene retrieval using two new components: 1) Neural Graphics Noise Analysis and 2) Smart Camera Movement Module.
- Experimental results are promising.

Weakness:
- There are some missing comparisons with recent methods, e.g., TIGER or N2F2 (but these are not open-sourced, so a direct comparison is not feasible). The authors provided a detailed description of the difference between these recent methods.
- unclear sensitivity to the camera initialization.

Justification:
- The paper receive consistent positive feedback from four reviewers. Three of them rate this work as "8: accept, good paper". The rebuttal further clarifies the remaining issues raised by the reviewers. All reviewers are satisfied with the responses. The AC agrees with the reviewers that this is a solid work and recommends to accept.

**Additional Comments On Reviewer Discussion:**

As summarized in the authors' response to reviewers, overall the reviewers are positive about this work.

The main discussions and changes are
- The authors updated section 2.4 to provide additional discussions with recent works.
- In Appendix A.4.1, the authors explain the camera initialization and convergence behavior.
- Include visualizations of retrieval results based on the prompt types suggested by reviewers in scenes of more complexity.

Overall, all reviewers agree that the concerns have been adequately addressed.

---

### Decision · Program_Chairs · 2025-01-22

Accept (Spotlight)